

# A reactivity continuum of particulate organic matter in a global ocean biogeochemical model

Olivier Aumont[1], Marco van Hulten[2], Matthieu Roy-Barman[2], Jean-Claude Dutay[2], Christian Éthé[1], and Marion Gehlen[2]

[1]Laboratoire d'Océanographie et du Climat: Expérimentations et Approches Numériques (LOCEAN), IPSL, 4 Place Jussieu, 75005 Paris, France
[2]Laboratoire des Sciences du Climat et de l'Environnement (LSCE), IPSL, CEA–Orme des Merisiers, 91191 Gif-sur-Yvette, France

*Correspondence to:* Olivier Aumont
(olivier.aumont@ird.fr)

**Abstract.** The marine biological carbon pump is dominated by the vertical transfer of Particulate Organic Carbon (POC) from the surface ocean to its interior. The efficiency of this transfer plays an important role in controlling the amount of atmospheric carbon that is sequestered in the ocean. Furthermore, the abundance and composition of POC is critical for the removal of numerous trace elements by scavenging, a number of which such as iron are

essential for the growth of marine organisms, including phytoplankton. Observations and laboratory experiments have shown that POC is composed of numerous organic compounds that can have very different reactivities. Yet, this variable reactivity of POC has never been extensively considered, especially in modeling studies. Here, we introduced in the global ocean biogeochemical model NEMO-PISCES a description of the variable composition of POC based on the theoretical Reactivity Continuum Model proposed by (Boudreau and Ruddick, 1991). Our model experiments

show that accounting for a variable lability of POC increases POC concentrations in the ocean's interior by one to two orders of magnitude. This increase is mainly the consequence of a better preservation of small particles that sink slowly from the surface. Comparison with observations is significantly improved both in abundance and in size distribution. Furthermore, the amount of carbon that reaches the sediments is increased by more than a factor of two, which is in better agreement with global estimates of the sediment oxygen demand. The impact on the

major macro-nutrients (nitrate and phosphate) remains modest. However, iron (Fe) distribution is strongly altered, especially in the upper mesopelagic zone as a result of more intense scavenging: Vertical gradients in Fe are milder in the upper ocean which appears to be closer to observations. Thus, our study shows that the variable lability of POC can play a critical role in the marine biogeochemical cycles which advocates for more dedicated in situ and laboratory experiments.





## 1 Introduction

The biological carbon pump regulates the atmospheric $CO_2$ levels by transferring large amounts of organic carbon produced by phytoplankton photosynthesis in the upper ocean to the deep interior (e.g., Falkowski et al., 1998). Transport to the ocean's interior occurs through sinking of organic particles, physical mixing and transport of both particulate and dissolved organic carbon (respectively POC and DOC), and vertical migrations by zooplankton. Of the organic carbon produced in the euphotic zone, only a small fraction escapes recycling in the upper part of the water column and is exported downwards being sequestered away from the atmosphere. It is suggested that a small modification of the recycling efficiency in the upper ocean can have a significant impact on atmospheric $CO_2$ (Kwon et al., 2009). The magnitude of the biological carbon pump is predicted to decrease in response to climate change (Steinacher et al., 2010; Bopp et al., 2013) resulting in a less efficient storage of carbon in the ocean.

POC fluxes and concentrations exhibit the strongest gradient in the mesopelagic zone which is the part of the ocean located between the bottom of the euphotic zone and about 1000 m. They decrease strongly with depth as particles are being fragmented, consumed and respired by zooplankton and bacteria. Numerous studies have attempted to describe the attenuation of the POC fluxes using simple relationships, the most commonly used being either power law functions (Martin et al., 1987) or exponential functions (Suess, 1980). In the power law function, the rate at which POC flux decreases with depth is controlled by the exponent $b$. In the exponential function, this is determined by the remineralisation length scale $z^\star$. In many model studies, either $b$ or $z^\star$ were kept constant to a specified uniform value (e.g., Collins et al., 2011; Hauck et al., 2013; Ilyina et al., 2013).

Yet, significant variations of these parameters have been evidenced using deep-sea sediment traps and particle imaging (Lamborg et al., 2008; Henson et al., 2012a; Guidi et al., 2015). For instance, the exponent $b$ has been shown to vary between about 0.4 and 1.75 (Guidi et al., 2015). Furthermore, models that use the simple power-law or exponential relationships fail in representing correctly both carbon fluxes and POC concentrations (Dutay et al., 2009, 2015). POC concentrations tend to be strongly underestimated in the deep ocean, which suggests that an excessive loss of POC is predicted in the mesopelagic zone.

Several factors have been proposed to explain the variability in the vertical profile of organic carbon remineralisation. Probably, the most popular of these factors is the 'ballast hypothesis' which stems from the strong correlation between POC fluxes and the load and composition in minerals (biogenic silica, calcium carbonate and lithogenic materials) found in the deep sediment traps (Francois et al., 2002; Klaas and Archer, 2002; Armstrong et al., 2001). In general, a high preservation efficiency of POC is related to a high flux of calcium carbonate whereas no correlation has been found with biogenic silica. Calcium carbonate may either act as an efficient ballast for particles as a result of its high density or provides some protection against degradation by heterotrophic organisms (Francois et al., 2002; Klaas and Archer, 2002).

The 'ballast hypothesis' is currently debated as no coherent relationship has been found between the sinking speed of particles and the load in minerals (Lee et al., 2009) and the correlation between calcium carbonate and



the preservation efficiency of POC in the mesopelagic domain may not be that significant (Henson et al., 2012a). Despite the lack of consensus on the 'ballast hypothesis', many biogeochemical models currently describe POC fluxes based on that hypothesis (e.g., Moore et al., 2004; Dunne et al., 2012). The other factors that may impact on the remineralisation of POC in the mesopelagic zone include the ecosystem structure (Boyd et al., 1999), temperature (Matsumoto, 2007; Marsay et al., 2015), oxygen (Devol and Hartnett, 2001) and the pressure effect on bacterial activity (Tamburini et al., 2003; Grossart and Gust, 2009).

The composition of POC is an alternative factor that has not been extensively considered in previous studies on POC fluxes. Organic matter is composed of numerous compounds that can have very different reactivities. The variable lability of organic matter has been evidenced for dissolved organic carbon both in the ocean (Amon and Benner, 1994; Hansell, 2013; Benner and Amon, 2015) and in lakes (Koehler et al., 2012) and for organic carbon in the sediments (Middelburg, 1989; Boudreau and Ruddick, 1991). Several studies have also shown that POC is a complex mixture of compounds with different labilities (Sempéré et al., 2000; Panagiotopoulos et al., 2002; Benner and Amon, 2015). The labile fractions of POC can be remineralised in the upper ocean and in the mesopelagic domain, the more refractory fractions being exported to the deep ocean. That could explain the variability in the transfer efficiency of POC to the deep ocean and the slow vertical decrease of POC in the deep ocean (Marsay et al., 2015). With the notable exception of (Sempéré et al., 2000), the inhomogeneous reactivity of POC has not been taken into account in marine biogeochemical modeling studies, the vast majority of the models using a single uniform decay rate for the whole particulate organic pool.

Here, we explore how a variable reactivity of POC affects the vertical export of carbon by sinking particles, the POC concentrations in the ocean's interior and the marine carbon cycle overall. Several theoretical frameworks have been proposed to describe the variable lability of POC (e.g., Boudreau and Ruddick, 1991; Rothman and Forney, 2007; Vähätalo et al., 2010). We have implemented in the coupled ocean physical and biogeochemical model NEMO-PISCES (Aumont et al., 2015) the reactivity continuum model proposed by Boudreau and Ruddick (1991). Using that model, we analyze the consequences of a variable lability of organic matter on the spatial distribution of POC and on the marine carbon cycle.

## 2 Methods

### 2.1 The biogeochemical global ocean model

In this study, we use the biogeochemical model PISCES-v2 (Aumont et al., 2015) as part of the modelling framework NEMO (Madec, 2008). PISCES, presented in Figure 1, has been employed for many other studies concerning trace metals, as well as large-scale ocean biogeochemistry (e.g. Gehlen et al., 2007; Arsouze et al., 2009; Dutay et al., 2009; Tagliabue et al., 2010; Van Hulten et al., 2014). The model simulates the biogeochemical sources and sinks of 24 prognostic tracers, including five limiting nutrients (Fe, $PO_4$, $Si(OH)_4$, $NO_3$ and $NH_4$) Two phytoplankton groups (nanophytoplankton and diatoms) and two zooplankton size-classes (microzooplankton and mesozooplankton) are





represented in PISCES. Fixed Redfield ratios are prescribed for N and P to values proposed by Takahashi et al.
(1985) whereas the ratios of both Fe and Si to C are explicitly modeled. There are three non-living compartments:
semi- labile dissolved organic carbon and two size-classes of particulate organic carbon.

As this study focuses on POC, we briefly describe the standard parameterisation used in PISCES to model the
evolution of this carbon pool. For more information, the reader is referred to the complete description of PISCES-v2
presented in Aumont et al. (2015). The two size-classes of POC correspond to small slow-sinking POC and big
fast-sinking POC. Both pools of POC are produced by the aggregation of phytoplankton, the mortality of both
phytoplankton and zooplankton, the egestion of fecal pellets and the coagulation of dissolved organic carbon. It
is consumed by zooplankton feeding and remineralization by heterotrophic bacteria. Bacteria are not explicitly
modeled and a first order kinetics is assumed to represent the impact of the bacterial activity on POC. The specific
degradation rate $\lambda^{\star}_{POC}$ is the same for both size-classes of POC and depends on temperature with a $Q_{10}$ of 1.9. In
this version of PISCES, the sinking speeds of both POC compartments are supposed constant and uniform, and are
set to $2\,\mathrm{m\,d^{-1}}$ for small POC and to $50\,\mathrm{m\,d^{-1}}$ for big POC.

This model is embedded in the ORCA2-LIM configuration of NEMO (Madec, 2008). The spatial resolution is about
$2°$ by $2°\cos(\phi)$ (where $\phi$ is the latitude) with an increased meridional resolution to $0.5°$ in the equatorial domain.
The model has 30 vertical layers, with an increased vertical thickness from 10 m at the surface to 500 m at 5000
m. Representation of the topography is based on the partial-step thicknesses (Barnier et al., 2006). Lateral mixing
along isopycnal surfaces is performed both on tracers and momentum (Lengaigne et al., 2003). The parameterisation
of Gent and McWilliams (1990) is applied poleward of $10°$ to represent the effects of non-resolved mesoscale eddies.
Vertical mixing is modeled using the turbulent kinetic energy (TKE) scheme of Gaspar et al. (1990), as modified by
Madec (2008). The dynamics used to drive PISCES is identical to that used in Aumont et al. (2015).

## 2.2   The Reactivity Continuum Model

In the standard version of PISCES, as commonly in biogeochemical models, the decomposition of POC is described
following first-order kinetics. Most frequently, a single constant decay rate $k$ is used which implies that all components
of organic matter degrade at the same rate. However, organic matter is a complex mixture of compounds of varying
origin and different reactivities. Therefore, a single decay constant often fails to represent the observed degradation
kinetics of organic matter (e.g., Middelburg et al., 1993; Del Giorgio and Davis, 2002). To overcome that problem,
several models have been proposed (Arndt et al., 2013). In multi-G models, organic matter is split into discrete pools
with different labilities (e.g., Ogura, 1975). The power model describes the mean first-order decay coefficient $\bar{k}$ as a
power function of the apparent age of organic matter $t$ (Middelburg, 1989). The Reactivity Continuum (RC) models
extend the formalism introduced in the multi-G models (Aris and Gavalas, 1966). Instead of a few discrete pools of
different reactivities, the RC models use an infinite number of pools characterised by a continuous distribution of
reactivities in the substrate. Several functions have been proposed to model this distribution: A gamma distribution
(Boudreau and Ruddick, 1991), a beta function (Vähätalo et al., 2010), and a lognormal function (Rothman and





Forney, 2007). The power model and the RC models have been shown to be particular cases of a more general formulation called *q-theory* (Bosatta and Ågren, 1991; Bosatta and Ågren, 1995).

In this study, we choose to describe the decomposition of particulate organic carbon according to the reactivity continuum model. As in Boudreau and Ruddick (1991), the distribution of reactivities of newly produced POC is represented by a gamma distribution:

$$g(k,0) = \frac{g_0 k^{\nu-1} e^{-ak}}{\Gamma(\nu)}, \tag{1}$$

5 where $\nu$ describes the shape of the distribution near $k = 0$, and $a$ is the average life time of the more reactive components of POC. The corresponding cumulative distribution function (CDF) is defined as:

$$\mathcal{G}(k,0) = \frac{\gamma(\nu, ak)}{\Gamma(\nu)} = \frac{\int_0^{ak} x^{\nu-1} e^{-x} dx}{\int_0^{\infty} x^{\nu-1} e^{-x} dx} \tag{2}$$

where $\gamma(\nu, ak)$ is the lower incomplete gamma function. To get the time-evolved distribution, we assume first-order decay for each lability class:

10 $$g(k,t) = \frac{g_0 k^{\nu-1} e^{-(a+f(T)t)k}}{\Gamma(\nu)}, \tag{3}$$

where $f(T)$ is a function of temperature. As in the standard version of PISCES, the dependency to temperature corresponds to a $Q_{10}$ of 1.9. In that equation, the effect of temperature on the distribution is equivalent to defining a pseudo time variable $t^\star = f(T)t$.

In a closed system with a constant temperature $T$, the mean remineralisation rate of POC decreases with time 15 and is described by a simple function of the pseudo time variable $t^\star$:

$$\frac{\mathrm{d}POC}{\mathrm{d}t^\star} = -\frac{\nu}{a+t^\star} POC. \tag{4}$$

Unfortunately, in open systems such as the ocean, the RC model cannot be used in its continuous form since transport, production and consumption of organic matter alter the shape of the distribution. As a consequence, this distribution can significantly deviate from the initial gamma distribution. An option would be to model the 20 moments of the reactivity distribution following an approach similar to what is done for instance in the atmosphere for aerosols (Milbrandt and Yau, 2005) or in the ocean for traits (Merico et al., 2009). However, the set of moment equations should be closed using a moment closure approximation to truncate the system at a certain order. Since the distribution can deviate to a non-specific form, the number of moments that should be tracked becomes very large (over ten moments based on some preliminary tests we performed) which makes that method not computationally 25 efficient. Instead, the distribution is discretised by separating both small and large POC into a finite number of pools having degradation constants that are equally spaced in the natural logarithmic transform of the reactivity space. In this study, we have arbitrarily set the smallest and largest degradation rate constants of the reactivity space to respectively $\bar{\lambda}/1000$ and $10\bar{\lambda}$, where $\bar{\lambda}$ is the mean degradation rate of freshly produced POC. Each degradation rate





constant is computed as:

$$k_i = \frac{1}{1000}(1 \times 10^4)^{\frac{i-1}{n-2}} \quad \text{for} \quad i = 1, n-1 \tag{5}$$

The fraction $\mathcal{G}(\bar{k}_i, 0)$ and the mean degradation rate $\bar{k}_i$ of POC having degradation constants between $k_i$ and $k_{i+1}$ are:

$$
\begin{aligned}
\mathcal{G}(\bar{k}_i, 0) &= \frac{\gamma(\nu, ak_{i+1}) - \gamma(\nu, ak_i)}{\Gamma(\nu)} \\
\bar{k}_i &= \frac{\gamma(\nu+1, ak_{i+1}) - \gamma(\nu+1, ak_i)}{\Gamma(\nu)}
\end{aligned} \tag{6}
$$

We thus use a multi-G model to simulate the reactivity continuum. An identical approach has been used by Dale

et al. (2015) to model the degradation of the organic matter in the sediments. Based on experiments performed with a 1-D model, the number of pools has been set to 15 for both small and big POC which results in a less than $1\,\%$ error relative to the exact solution. Furthermore, for the sake of simplicity, we assumed that the shape factor $\nu$ is equal to 1.

An explicit representation of the reactivity of the organic matter would require to have 30 distinct pools, which

would more than double the number of variables in PISCES (24 tracers including big and small POC). This would thus considerably increase the computing cost of the model. To overcome that problem, we made a rather strong assumption: We postulated that the lability distribution of POC is insensitive to ocean transport and is only modified by sinking, by the biological sources and sinks, and by vertical mixing in the mixed layer. This assumption is further discussed in the discussion section of this study. The problem is thus reduced to a 1-D framework and the

vertical distribution of each pool can be iteratively solved starting from the base of the mixed layer. In a lagrangian framework, the system to be solved is:

$$
\begin{aligned}
\frac{\mathrm{d}z}{\mathrm{d}t} &= w_{\text{POC}} \\
\frac{\mathrm{d}\mathcal{G}(k,t)POC}{\mathrm{d}t} &= \mathcal{G}(k,0)P_{\text{POC}} - \mathcal{G}(k,t)S_{\text{POC}} \\
&\quad - kf(T)\mathcal{G}(k,t)POC \\
\mathcal{G}(k,t) &= \mathcal{G}_{\text{mxl}}(k,t) \quad \text{if} \quad z = z_{\text{mxl}},
\end{aligned} \tag{7}
$$

where $z$ is positive downwards, $P_{\text{POC}}$ denotes the production of particulate organic carbon, $S_{\text{POC}}$ is the sink of POC, $w_{\text{POC}}$ the settling velocity of POC, $z_{\text{mxl}}$ the depth of the bottom of the mixed layer, and $\mathcal{G}(k,t)$ corresponds to the

mass fraction of POC with a decay rate $k$ at time $t$. Assuming constant sources and sinks over each grid cell, this system can be solved analytically. The distribution is then computed at each time step and a mean $\bar{k}$ is inferred from that distribution which is then used in PISCES to model the decomposition of POC. Of course, this computation is done independently for small and big POC.

The solution of system (7) requires to know the distribution at the bottom of the mixed layer $G_{\text{mxl}}(k,t)$. In the

mixed layer, ocean dynamics, especially vertical mixing, cannot be neglected. Since vertical mixing is strong, tracers





in the mixed layer, including the reactivity distribution, can be considered homogeneous. Using that assumption, the mean reactivity distribution can be computed as:

$$\mathcal{G}_{\mathrm{mxl}}(k,t) = \frac{\int_0^{z_{\mathrm{mxl}}} \mathcal{G}(k,0) P_{\mathrm{POC}} \mathrm{d}z}{\int_0^{z_{\mathrm{mxl}}} (kPOC + S_{\mathrm{POC}}) \mathrm{d}z + w_{\mathrm{POC}} POC(z = z_{\mathrm{mxl}})}. \tag{8}$$

The lability parameterisation introduces an extra cost of about 20 %, but it depends of course on the number of lability pools.

### 2.3 Model experiments

The model setup used in this study is exactly identical to that described in Aumont et al. (2015) except for the modifications made on POC described above. All modeled experiments presented here have been initialised from the quasi steady-state simulation presented in Aumont et al. (2015). Two different simulations have been performed: A control experiment which is based on the standard version of PISCES (noRC) and a second experiment in which the variable lability of POC based on the reactivity continuum model is used (RC). In the RC experiment, the rate parameter $a$ has been prescribed so that the global mean first order degradation rate in the top 50 m of the ocean is identical to the value prescribed in the standard model configuration (the initial degradation rate $k = \nu/a$). Each experiment has been run for 1000 years. After that duration, POC distribution was in an approximate steady state. In Table 1, we present an overview of the two simulations.

### 2.4 Observations

To test the model performance, we use measurements of POC concentrations performed in the Atlantic and Pacific Oceans, whose stations are presented in Figure 2. They relatively sparsely cover these oceans. Unfortunately, to our knowledge, there are no particle data available in the other ocean basins. Since the POC concentration is an ancillary parameter of GEOTRACES, the number of POC data should considerably increase in the near future. In the Intermediate Data Product (IDP2014) of GEOTRACES released recently (Mawji et al., 2015), only the GA03 transect (the red crosses displayed in Figure 2 in the North Atlantic Ocean) includes POC observations.

Observed POC fluxes are from Dunne et al. (2005); Gehlen et al. (2006) and Le Moigne (2013). In these datasets, data have been obtained from sediment traps and/or $^{234}$Th. The deep sediment trap data in Gehlen et al. (2006) have not been Th-corrected. In addition to these data, we also use a global distribution of oxygen fluxes at the sediment-water interface (Jahnke, 1996). Assuming full oxic remineralisation of POC, and using the value of the Redfield ratio of PISCES, we computed from the oxygen fluxes the equivalent fluxes of POC to the sediment.



## 3   Results

### 3.1   Concentrations of POC

Figures 3 and 4 present POC profiles in the Atlantic and Pacific Oceans, respectively. Both data and model results are displayed below 100 m since we focus on the fate of particles once they have left the productive upper zone. Furthermore, in the euphotic zone, a significant fraction of POC consists of living organisms such as phytoplankton and zooplankton which are out of the scope of this study. In the observations, POC concentrations range from about 0.4 to more than 2.5 µM in the upper part of the water column, the lowest values being found in the oligotrophic areas. In the upper part of the mesopelagic domain, between 100 m and about 500 m, POC steeply decreases to reach values that range between 0.05 and 0.15 µM. In the deep ocean, concentrations remain relatively constant and decrease only slowly to the bottom of the ocean. Elevated concentrations of POC can be observed in the deep ocean and correspond to nepheloid layers generated by the resuspension of sedimentary materials or by hydrothermal vents (such as in the oligotrophic Atlantic Ocean, Figure 3 c) (Lam et al., 2015a).

Our model experiments reproduce quite correctly the general shape of the observed profiles as shown in Figures 3 and 4. However, in the `noRC` model configuration, the decrease in POC is too steep and concentrations in the bottom part of the mesopelagic domain are strongly underestimated by at least a factor of 2. Furthermore, POC continues to strongly decrease in the deep ocean where modeled concentrations are at least an order of magnitude too low relative to the observations. The parameterisation of a variable lability of biogenic organic matter (`noRC` experiment) improves quite notably the agreement between the model and the observations, both in the mesopelagic domain and in the deep ocean. In particular, POC concentrations in the deep ocean are now comparable to the observed values and do not strongly decrease with depth. However, two main biases are still produced by the model. First, POC concentrations tend to be overestimated at around 100 m and are explained by the living biomass which suggests that PISCES overpredicts phytoplankton and zooplankton levels in the lower part and just below the euphotic zone. Second, elevated POC concentrations are not simulated in the deep ocean. Such result is not surprising since the model does represent neither the resuspension of sediments, nor the hydrothermal vents.

Figures 5 and 6 present the relative contribution of large POC to total POC. In the observations, this contribution does not exhibit any significant vertical trend. Despite a significant scatter, it remains more or less constant with depth at values that are generally comprised between 0.1 to 0.4. In the standard configuration of the model, the fraction of large organic particles steeply increases with depth in the upper part of the mesopelagic domain. Then, it remains relatively stable below 500 m to values around 0.7. Below the euphotic zone, small particles are rapidly removed from the water column due to their slow sinking speed associated to a relatively strong remineralisation. As a consequence, the contribution of large particles rises rapidly as POC sinks downward. Then, small particles are continuously produced by the degradation of large particles and a quasi steady-state is reached between this source and the loss from remineralisation. This means that small POC in the deep ocean as simulated in the `NoRC` experiment is locally produced. Thus, the organic materials in this size fraction of POC is very young and none of it





originates directly from production in the upper ocean. This result is in apparent contradiction with the radiocarbon
age of suspended POC (Druffel and Williams, 1990; McNichol and Aluwihare, 2007), even if alternative explanations
may explain the increasing apparent age of this pool with depth (McNichol and Aluwihare, 2007).

The vertical profiles of the relative contribution of large POC are notably impacted by the lability parameterisation
introduced in this study (Figures 5 and 6). In the RC experiment, this contribution does not significantly increase
with depth. Below the upper 200 m, the proportion of large POC stabilises and remains then almost constant to
values between 0.1 and 0.3. The simulated profiles thus fall in the observed range. When produced, a significant
fraction of POC is refractory and escapes rapid remineralisation in the upper ocean. This refractory component
builds up in small POC because of its slow sinking speed, which explains the dominant contribution by this pool in
the mesopelagic domain and in the deep ocean. In contrast with the NoRC experiment, a significant fraction of small
POC originates from the surface.

These results suggest a strong improvement of the representation of POC below the ocean surface, especially
for small POC. A more quantitative evaluation of the model experiments is presented in Table 2. In addition to
classical statistical indices (the correlation coefficient ($r$), the root mean square error (RMSE) and the bias (B)), two
additional performance indicators have been used as suggested in previous skill assessment studies (Allen et al., 2007;
Stow et al., 2009; Vichi and Masina, 2009): The Reliability Index (Leggett and Williams, 1981) and the Modelling
Efficiency (MEF) (Nash and Sutcliffe, 1970). The RI indicates the average ratio by which the model differs from
the observations. The RI should be close to one. The MEF measures how well the model predicts the observations
relative to the average of the observations. A value of one denotes a close match with the observations. A negative
value means that the average of the observations is a better predictor than the model.

The scores of the RC experiment are much better than those of the standard model, especially for the RI and the
MEF. In particular, the MEF is positive and relatively close to 1. Such result proves that the model performs better
than the average of the observations which is not the case of the model without the lability parameterisation. The
Bias is notably reduced. Such improved scores in all statistical indicators confirm the visual inspection of the vertical
profiles displayed in Figures 3 and 4. On the other hand, the correlation coefficient $r$ is, comparatively to the other
indicators, only modestly improved. This suggests that the spatial patterns of POC are not strongly impacted by
the new lability parameterisation.

## 3.2 Fluxes of POC

Table 3 shows the area-integrated carbon fluxes of the world ocean, according to our simulations and estimates
based on observations. The primary production predicted in all three simulations ranges from 41 to 52 PgC/yr and
lies within the observed estimates. The export fluxes from the euphotic zone are highest for the simulation without
lability, consistent with the higher global primary production rate. The lowest productivity and export out of the
euphotic zone are predicted by the RC experiment. Conversely, the latter experiment produces the highest export at
2000 m as well as a flux to the sediments that is more than twice as high as in the standard model configuration.





Such higher export in the deep ocean stems from the much higher concentrations of small POC in the deep ocean as shown in the previous section. Thus, the increase in export is explained by a higher abundance of small particles.

30  On the global scale, the relative contribution of small POC rises from almost 0 in the `NoRC` experiment to about 20% in the `RC` experiment. This significant contribution of small slowly-sinking particles to the export of carbon in the interior of the ocean is supported by recent observations (Durkin et al., 2015). The export at 2000 m, about 0.8 PgC/yr, is overestimated relative to recent estimates by Henson et al. (2012b) and Guidi et al. (2015), respectively 0.45 and 0.33 PgC/yr. Since the observed POC concentrations appear to be well reproduced by the model, this suggests that the sinking speeds of the biogenic organic particles might be overestimated. On the other hand, the predicted POC export to the deep sea sediments falls on the low end of the estimated range.

Figure 7 shows the spatial distribution of the annual mean fluxes of POC at 100 m and at 2000 m in the `NoRC` and `RC` experiments. At 100 m, fluxes display very similar spatial patterns. They span about two orders of magnitude

with low values in the oligotrophic subtropical gyres and much higher values in the high latitudes and in the eastern boundary upwelling systems, such as the Peru and the Benguela upwelling regions where they exceed $4 \, \mathrm{mmol\,C/m^2/s}$. The most noticeable difference between the two experiments is a weaker export in the Southern Ocean in the `RC` experiment. The horizontal patterns are qualitatively similar to those published by Henson et al. (2012b). However, our simulated export fluxes are on average larger which explains our larger global mean estimate, 8.1–9 versus 4

PgC/yr in Henson et al. (2012b) (see Table 3). Comparing to individual sediment traps data is rather challenging because they exhibit a large scatter. Since our focus is the fate of particles and the export of carbon in the mesopelagic domain and in the deep ocean, shallow export is not further discussed.

Fluxes at 2000 m are much lower than at the euphotic depth (Figures 7c and 7d). Their horizontal variability is also reduced since they span about 1 order of magnitude, ranging from around 0.1 to less than $1 \, \mathrm{mmol\,C/m^2/s}$. As

expected from the global mean fluxes (Table 3), values predicted in the `RC` experiment are higher than in the `NoRC` experiment. As already discussed, this increase in export is explained by the much larger contribution from small POC when the lability parameterisation is applied. Comparison with published horizontal distributions (Henson et al., 2012b; Guidi et al., 2015) shows quite similar horizontal patterns but suggests that our predicted fluxes are overestimated. This is confirmed by evaluation against individual sediment traps data shown in Figure 7, even

though these data are very sparse.

The benthic oxygen demand provides an indirect measurement of the biogenic organic carbon flux to the sediments, at least when anoxic processes are negligible, which is generally true in deep sea sediments (Archer et al., 2002). Figure 8 compares POC fluxes to the sediments derived from a global database of the benthic oxygen demand (Jahnke, 1996) with those from our model simulations. As opposed to the model without lability, the order of

magnitude and basic horizontal patterns predicted by the lability simulation compare well with the observations. Such agreement appears in apparent contradiction with the POC fluxes at 2000 m which were found to be quite too large.





## 4  Discussion

### 4.1  Model caveats

This study relies on the hypothesis that POC is made of various compounds with varying lability. Several studies based on observations support that hypothesis (e.g., Sempéré et al., 2000; Panagiotopoulos et al., 2002; Benner and Amon, 2015). To describe this variable lability of POC, we have made the rather strong assumption that the reactivity of newly produced POC follows a gamma distribution. Several motives explain this choice. Firstly, the degradation of POC in the sediments has been shown to be well described by the reactivity-continuum model based on a gamma-distribution (e.g., Boudreau and Ruddick, 1991; Marquardt et al., 2010; Wadham et al., 2012). Since organic matter in the sediments is deposited from the ocean, this supports the assumption that the reactivity of POC that reaches the sediments can be adequately modeled by a gamma distribution. Secondly, the gamma model

provides a complete quantitative description of the heterogeneous biodegradability of particulate organic matter which agrees with qualitative descriptions of degradation kinetics for POM (Amon and Benner, 1996). Thirdly, the assumption of a gamma distribution makes the mathematical handling quite convenient since the mean degradation rate constant can be easily computed (see Equation 4).

Several alternative expressions of the biodegradability distribution have been proposed in the literature: a beta

function (Vähätalo et al., 2010), a log-normal distribution (Rothman and Forney, 2007) and Gaussian and Weibull distributions (Burnham and Braun, 1999). Yet, despite their different mathematical expressions, all these models lead to quite similar mean kinetics for POC (Vähätalo et al., 2010; Boudreau et al., 2008).

A supplementary interesting aspect of the gamma model is that it can be described by only two parameters: the shape parameter $\nu$ and the rate parameter $a$. For comparison, a three-pool multi-G model requires five parameters.

In this study, we prescribed both parameters to be constant and uniform over the global ocean. For the sake of simplicity, $\nu$ has been set to 1. Then, the rate parameter $a$ has been computed to obtain a global mean first order degradation rate in the top 50 m of the ocean that is identical to the value prescribed in the standard model configuration (the initial degradation rate $k = \nu/a$). Yet, several studies have shown that this parameter can be quite variable and is rather comprised between 0.1 and 0.2, which is smaller than our prescribed value (Boudreau

and Ruddick, 1991; Boudreau et al., 2008; Koehler et al., 2012). However, most of these estimates are based on sedimentary materials which are already quite degraded. This may explain the low $\nu$ values which are characteristic of a large contribution of refractory compounds.

Furthermore, the temporal evolution of the total POC concentration can be expressed in a closed system as:

$$POC(t) = POC(0) \left( \frac{a}{a+t} \right)^{\nu} \tag{9}$$



where $POC(0)$ is the initial POC concentration. Assuming that POC is sinking with a constant sinking speed w, this equation can be rewritten as a function of depth:

$$POC(z) = POC(0) \left( \frac{aw}{aw + z} \right)^{\nu} \tag{10}$$

For depth $z$ much larger than $aw$, POC concentrations should vary vertically as $z^{-\nu}$, i.e. as an inverse power function of depth. From that analysis, $\nu$ is thus equivalent to $b$, the exponent used in the popular relationship proposed by Martin et al. (1987). Based on the analysis of sediment traps deployed at nine stations located in the Pacific Ocean, Martin et al. (1987) found that $b$ is on average equal to about 0.86 which is not very far from our prescribed value of 1. Such asymptotic behavior of the reactivity continuum formalism might appear as a limit. Indeed, several studies have shown that $b$ can be quite variable and can range from about 0.4 to more than 1.5 (Henson et al., 2012a; Guidi et al., 2015). However, the analytical profile presented in Equation 10 is valid only in a closed system with a homogeneous pool sinking at the same settling speed. In the ocean and in the model, POC is a mixture of materials sinking at different speeds and produced all along the water column. As a consequence, $b$ is not predicted to be constant and can significantly differ in the model from the analytical value of 1, as we will show in the next section.

Observations based on sedimentary materials and those from sediment traps seem to produce contradicting results concerning the order of magnitude of the shape coefficient $\nu$, about 0.1 versus about 1. As a sensitivity experiment, we have run the RC-model using a shape coefficient set to 0.16 as proposed by Middelburg (1989). The rate parameter $a$ has been prescribed so that the predicted mean remineralisation rate in the top 50 m of the ocean is equivalent to that in the `RC` experiment. The resulting vertical distribution of POC exhibits too weak vertical gradients in the upper ocean, as well as excessive concentrations in the deep ocean (not shown). This suggests that the contribution of refractory compounds in freshly produced organic matter is too high. The lower shape coefficients in the sediments may be explained by the continuous mixture between old materials coming from the surface and fresher organic matter supplied by the large particles and by the zooplankton activity in the water column. To test that hypothesis, we have computed from the RC-model the vertical structure of the shape coefficient corresponding to the lability distribution of total POC. The model predicts that $\nu$ decreases with depth from less than 0.9 in the upper ocean to about 0.45 at 4500m (Figure 9). At that depth, POC is a mixture of organic matter whose age ranges from 0 to about 6 years. Thus, our model supports the hypothesis that the relatively low values of the shape coefficient in the sediments can be produced by a heterogeneous age of the organic matter that is buried in the sediments.

The implementation of the reactivity continuum model relies on the very strong assumption that advection and diffusion do not affect the lability distribution of POC (except for diffusion in the mixed layer). This assumption may appear legitimate for large POC because of its large settling speed. However, for small POC, it may introduce large errors in the model behavior. To test the impact of that assumption, we performed a model experiment in which the lability classes of small POC are modeled as individual prognostic tracers. Large POC is represented using the simplified framework like in the experiment `RC`. Since this experiment is quite computationally intensive,





we discretise the lability space using 9 classes. This simulation has been run for 30 years and is compared to the equivalent simulation using the simplified lability parameterization with the same 9 lability classes.

Figure 10 shows the remineralisation rate and the POC concentrations computed using the nine prognostic POC classes versus the simplified lability model in log space. Both scatter plots displayed on Figure 10 suggest that the assumption on which we based the implementation of the reactivity continuum model does not introduce severe

biases in the solution predicted by the model. Differences remain relatively modest, except in regions which experience deep vertical mixing such as in the North Atlantic Ocean and in the Southern Ocean. Furthermore, the borders of the upwelling regions also exhibit significant biases, especially in remineralisation rates. Nevertheless, biases remain much smaller than the differences in POC concentrations between the `noRC` and `RC` experiments.

## 4.2  Spatial variations of remineralisation efficiency

Important spatial variations of the remineralisation efficiency in the mesopelagic domain have been evidenced using a combination of sediment traps data, underwater imaging systems, Th-derived fluxes and POC observations (e.g., Berelson, 2001; Lam et al., 2011; Guidi et al., 2015). These variations have been explained by differences in the

community distribution and composition (Boyd et al., 1999; Guidi et al., 2015), in the zooplankton activity in the mesopelagic domain (Stemmann et al., 2004; Robinson et al., 2010), in temperature (Marsay et al., 2015), in oxygen (Devol and Hartnett, 2001) and in bacterial activity in response to changes in pressure (Tamburini et al., 2003). The remineralization efficiency in the mesopelagic domain can be described by the remineralisation exponent $b$ used in the power-law function proposed by Martin et al. (1987). High values of $b$ denote intense shallow remineralisation,

whereas low values indicate efficient transfer to the deep ocean.

Several studies have attempted to estimate the regional distribution of $b$ at the global scale based on the analysis of global collections of *in situ* data (Henson et al., 2012a; Guidi et al., 2015; Marsay et al., 2015). Considerable differences exist between these different estimates. In Henson et al. (2012a) and Guidi et al. (2015), high values of $b$, generally above 1, are found in the high latitudes and in eastern boundary upwelling systems whereas lower values,

typically below 0.6, are estimated in the low latitudes, especially in the oligotrophic subtropical gyres. In Marsay et al. (2015), the spatial distribution of $b$ is completely reversed with high values in the subtropical values and much lower values in the high latitudes. A potential explanation to these apparently contradicting results is the depth range of the data that have been used to estimate $b$ (Marsay et al., 2015). Yet, Guidi et al. (2015) have used POC profiles observed with Underwater Video Profilers which have a high vertical resolution and are restricted to the top

1000m, similar to the vertical range analyzed in Marsay et al. (2015).

We have shown that the varying lability of the organic matter strongly changes the distribution of POC both in the mesopelagic zone and in the deep ocean. As a consequence, the remineralisation efficiency below the euphotic zone should be significantly impacted when accounting for a variable reactivity of POC. Figure 11 displays the anomalies of $b$ relative to the median value of that coefficient, both in the `NoRC` and in the `RC` experiments. The first

major difference between the two experiments is the median $b$ value. In the `NoRC` experiment, it is equal to 0.87, very



close to the estimate proposed by Martin et al. (1987) whereas it is lower, around 0.7, in the RC experiment. A lower coefficient in the latter experiment is not surprising as we showed that accounting for a variable lability leads to a better preservation of POC, especially of small particles, in the mesopelagic and deep domains. This median value of $b$, despite being lower, remains higher than in previous recent estimates proposed by Henson et al. (2012a) and

Guidi et al. (2015), respectively 0.54 and 0.64. The mean predicted $b$ coefficients are lower than the median values, 0.76 and 0.61 in the NoRC and RC experiment respectively. The predicted mean value in the RC experiment is close to the recent data-based estimates of 0.64 (Henson et al., 2012a; Guidi et al., 2015).

The spatial variations of $b$ predicted in both experiments are relatively similar. High values are simulated in the equatorial and eastern boundary upwelling systems and in the mid latitudes, where annual mean productivity is maximum. In contrast, the less productive areas of the low latitudes and the Arctic ocean are characterized by relatively low values of $b$. Our simulated distributions of $b$ do not compare very favorably with previous estimates (Henson et al., 2012a; Guidi et al., 2015; Marsay et al., 2015), despite some qualitative common features may be found with the first two quoted studies. However, as stated above, considerable disagreements exist between these

estimates which makes the assessment of our different model configurations quite challenging.

The largest difference between the two experiments is simulated in the central part of the subtropical oligotrophic areas. The NoRC experiment predicts high values of $b$ whereas the RC experiment predicts $b$ to be lower than the median value. Thus, accounting for a variable lability locally increases the efficiency of the transfer of carbon through the mesopelagic domain in these regions. The combination of a high temperature in the upper ocean and of a thick

productive zone (typically 200m) drives an efficient and rapid remineralisation of the labile fraction of organic matter near the surface, leaving the more refractory compounds for export to the interior of the ocean.

Except for the central subtropical gyres, both simulations exhibit similar spatial patterns. This suggests that a variable lability of the organic matter does not significantly change the spatial variations of the efficiency of remineralisation in the mesopelogic domain over most of the ocean. The spatial variability in our model is driven by

other mechanisms. Zooplankton activity in the interior of the ocean plays a major role and explains the simulated high $b$ values in productive areas of the ocean. Intense export out of the upper ocean sustains a high zooplankton biomass in the upper mesopelagic domain which very efficiently removes large and fast-sinking particles as suggested in previous studies (Stemmann et al., 2004; Iversen and Ploug, 2010; Jackson and Checkley Jr, 2011). Other factors such as temperature and the particle composition (the relative abundance of small and large particles) play a more

modest role. For instance, low $b$ coefficients in the Arctic Ocean are explained by a very low temperature and as stated above, the interplay between temperature and the variable lability of POC produces the differences between our two experiments in the subtropical gyres.

However, such finding relies on the assumption that the lability distribution of the freshly produced organic matter is spatially and temporally uniform. A different assumption would obviously change this finding. In particular, this

distribution may depend on the structure of the ecosystem in the euphotic zone. Biochemical analysis of the surface particulate organic matter have shown that its composition varies both in time and space as a result of differences




in the phytoplankton and zooplankton species and interactions (Tegelaar et al., 1989; Kiriakoulakis et al., 2001; Lee et al., 2004; Mayzaud et al., 2014). Thus, the lability distribution of POM is unlikely to be constant both in time and space. For instance, diatoms dominated ecosystems in the high latitudes are characterized by strong

seasonal blooms, exporting efficiently organic matter out of the euphotic zone. However, the transfer of this organic matter through the mesopelagic domain is quite inefficient, with only s small fraction of POM reaching the deep ocean (Lam et al., 2011; Mayor et al., 2012). Building on the distribution of Chl among phytoplankton size-classes developed by Uitz et al. (2008), Guidi et al. (2015) have also shown that the composition of the phytoplankton ecosystem partly controls the remineralisation efficiency below the euphotic zone. A potential explanation for this

low transfer efficiency in the mesopelagic domain in diatoms-dominated ecosystems is the large contribution of labile organic matter. This would translate in our Reactivity-Continuum formalism to a high value (above 1) of the shape coefficient $\nu$ (Arndt et al., 2013). Unfortunately, more data are needed to derive any general relationship or trend that would be applicable in a global model such as PISCES.

### 4.3  Impact of the lability of POC on the tracers distributions

A major effect of our variable lability parameterisation is to decrease the efficiency of remineralisation in the mesopelagic domain, especially for the small particles. Indeed, the transfer of organic matter to the deep ocean is strongly enhanced (see Table 3). As a consequence, the RC experiment should predict an enhanced sequestration of both carbon and nutrients in the lower part of the mesopelagic domain and in the deep ocean at the expense of the upper ocean and the upper mesopelagic domain. This results in a strong decrease by about 20% in primary pro-

ductivity (Table 3) which is more strongly limited by both micro- and macro-nutrients. The impact of the enhanced sequestration on the nutrient distributions (nitrate, phosphate, and silicate) remains generally modest (not shown). For instance, nitrate concentrations are decreased by on average $0.4\,\mu mol\,L^{-1}$ and at most $2.5\,\mu mol\,L^{-1}$ in the upper 200m. In the deep ocean, they are increased by on average about $0.08\,\mu mol\,L^{-1}$ and a maximum of $2\,\mu mol\,L^{-1}$. The impact on the $O_2$ distribution is larger, especially in the deep ocean where the enhanced export of organic matter

drives a more intense consumption. In the deep Pacific and Indian oceans, $O_2$ concentrations are decreased by on average $23\,\mu mol\,L^{-1}$.

The largest impact of the Reactivity-Continuum formalism is predicted on the iron distribution. Figure 12 shows the consequences of this formalism on the zonally averaged distribution of iron in the Pacific Ocean. The most striking feature is the strong decrease in dissolved Fe in the mesopelagic zone. The depth of the maximum anomaly is variable

and is roughly located close to the isoline denoting an iron concentration of $0.55\,nmol\,Fe/l$. Two mechanisms explain this subsurface decrease in iron. First, a less efficient remineralisation of biogenic iron in the mesopelagic zone decreases the release of dissolved iron. Second, the much larger POC concentrations in this zone drives a more intense scavenging of dissolved iron. This explains the depth of the maximum anomaly: The concentration of ligands in PISCES is set to $0.6\,nmol\,Fe/L$ everywhere (Aumont et al., 2015). Iron is strongly complexed by these ligands

and thus escapes scavenging by particles (Liu and Millero, 2002; Gledhill and Buck, 2012). When the dissolved





Fe concentration is getting close to the ligands concentration, the non-complexed free dissolved iron concentration increases rapidly just as scavenging does.

Figure 12 also displays a comparison between the simulated iron profiles of both experiments and a compilation of iron observations (Tagliabue et al., 2012). The analysis is restricted to the Pacific ocean but similar results are obtained in the other ocean basins. In both experiments, iron concentrations appear to be underestimated (nmol Fe/L) relative to the observations, especially in the `RC` experiment. The vertical increase in iron appears to be too steep in the `NoRC` experiment between 100 m and 400 m. As a consequence of the lower iron concentrations in the subsurface (see left panel of Figure 12), the vertical increase in iron in the `RC` experiment is weaker and is more similar to the observations. Yet, the iron concentrations are quite significantly underestimated in that experiment, by about 0.15 nmol Fe/L. This suggests that loss due to scavenging by particles is too intense. Such deficiency is not really surprising since the scavenging rate in the standard model has been adjusted with POC concentrations that are strongly underestimated (see Section 3.1). Consequently, this rate has been given a high value which has not been changed in the `RC` experiment despite much higher POC concentrations.

## 5 Conclusions

The lability of the marine organic matter has been extensively studied in the sediments (e.g., Middelburg, 1989; Boudreau and Ruddick, 1991; Arndt et al., 2013) and in the ocean for the dissolved component (e.g;, Amon and Benner, 1994; Hansell, 2013; Benner and Amon, 2015). The reactivity of POC in the ocean has received much less attention and has been mainly investigated through its interactions with lithogenic and biogenic inorganic particles, i.e. the 'ballast hypothesis' (e.g., Armstrong et al., 2001; Klaas and Archer, 2002; Iversen and Ploug, 2010). In this paper, we hypothesize that POC is composed of a mixture of compounds with varying lability. We propose to apply the Reactivity-Continuum model proposed by Boudreau and Ruddick (1991) to study the impact a variable lability might have on the POC distribution in the ocean and on the marine biogeochemical cycles. We describe an efficient parameterisation that has been embedded in the PISCES biogeochemical model (Aumont et al., 2015) and that is analyzed at the global scale using a coarse-resolution global configuration of NEMO.

A variable reactivity of POC leads to a large increase in concentrations in the mesopelagic domain and in the deep ocean, where concentrations are larger by about one to two orders of magnitude. This increase is explained mainly by a better preservation of small particles that sink slowly to the bottom of the ocean. The simulated vertical profiles of POC and the relative contribution of small particles to total POC are in much better agreement with the observations. Despite this large impact, the consequences on the nutrients distribution and on the export production of carbon out of the euphotic zone remain relatively small. Nevertheless, iron vertical gradients in the upper ocean are significantly reduced as a result of a stronger loss due to scavenging. Furthermore, the sequestration of carbon in the deep ocean is increased by about 55% relative to a model with a constant and uniform lability.





Previous modeling studies have shown that biogeochemical models simulate POC distributions that exhibit significant deficiencies which compromise their value when studying trace elements (Dutay et al., 2009, 2015). Here, we show that lability of POC may represent an explanation to these deficiencies. However, large uncertainties remain. In particular, we have used a gamma function to describe the lability distribution in POC as well as uniform and constant shape and reaction coefficients in that gamma function. Unfortunately, observations and experiments are currently insufficient to better constraint these uncertainties. We thus strongly advocate for more dedicated studies to better characterize the nature and the lability of POC, as well as its long term evolution when this organic matter ages in the water column.

## 6 Code availability

The source code of the NEMO model, including Pisces-v2 can be found at http://www.nemo-ocean.eu/. The official version with the lability parameterisation is available as a svn development branch called `dev_r6270_PISCES_QUOTA`. Note that many additional modifications of the standard version of Pisces-v2 are also available in that version of Pisces.

*Acknowledgements.* The help received from the Ferret community, and especially Ryo Furue, Patrick Brockman, Ansley Manke and Sarah Tavernel concerning visualisation and related technicalities is strongly appreciated. This work benefited from fruitful discussions with Laurent Bopp, Laurent Mémery and Alessandro Tagliabue. The funding of this work was through the European Project H2020 CRESCENDO.




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





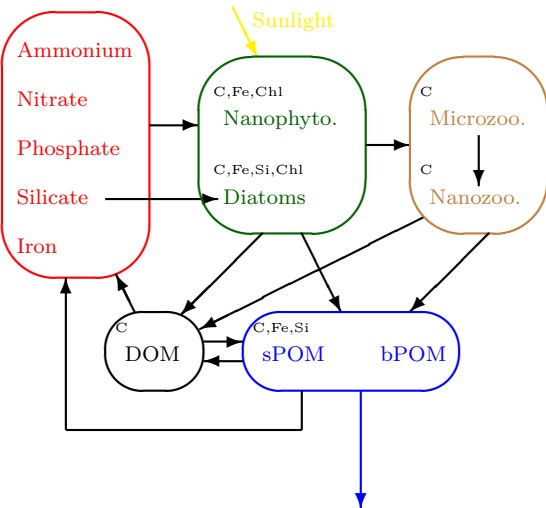

**Figure 1.** The biogeochemical model PISCES. DOM stands for dissolved, and POM for particulate organic matter.

Vichi, M. and Masina, S.: Skill assessment of the PELAGOS global ocean biogeochemistry model over the period 1980?2000, Biogeosciences, 6, 2333–2353, 2009.

Wadham, J. L., Arndt, S., Tulaczyk, S., Stibal, M., Tranter, M., Telling, J., Lis, G. P., Lawson, E., Ridgwell, A., Dubnick, A., Sharp, M. J., Anesio, A. M., and Butler, C. E. H.: Potential methane reservoirs beneath Antarctica, Nature, 488, 633–637, doi:10.1038/nature11374, 2012.





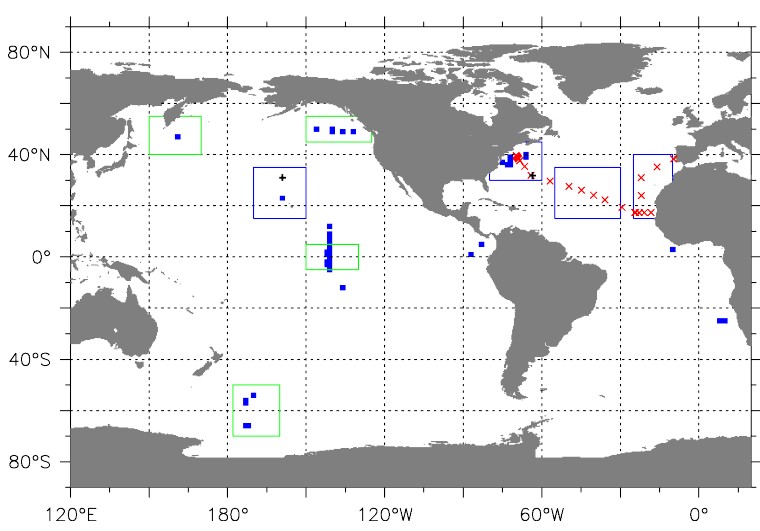

**Figure 2.** The particle data currently available. The blue squares present the stations of the data collection by Lam et al. (2011). The red crosses show the station coordinates of Hayes et al. (2015); Lam et al. (2015b). The two black plus signs north of Hawaii and east of Bermuda are the stations of Druffel et al. (1992). The rectangles define the regions of the profiles plotted in Section 3 (green: up to 1 km depth, blue: up to 6 km depth).





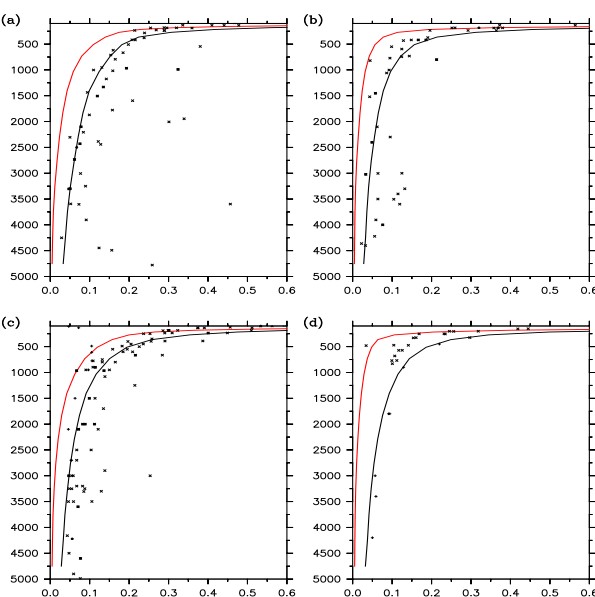

**Figure 3.** Modelled and observed total POC concentrations (µM) in different regions of the ocean: **(a)** western, **(b)** oligotrophic and **(c)** eastern North Atlantic Ocean (Lam et al., 2015b); **(d)** Hawaii region. The continuous lines are concentrations averaged over the region marked by the blue rectangles on the map of Figure 2: without (in red) and with the reactive continuum (RC) parameterization (in black). The black speckles are observations in the respective regions from Druffel et al. (1992); Lam et al. (2015b).





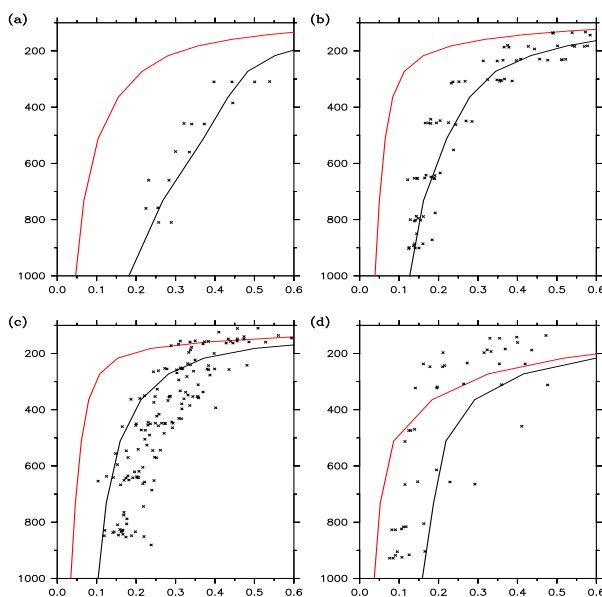

**Figure 4.** Same as figure 3 in additional regions of the ocean: **(a)** northwest, **(b)** northeast, **(c)** central and **(d)** southern Pacific Ocean. Observations (black speckles) are from Lam et al. (2011).





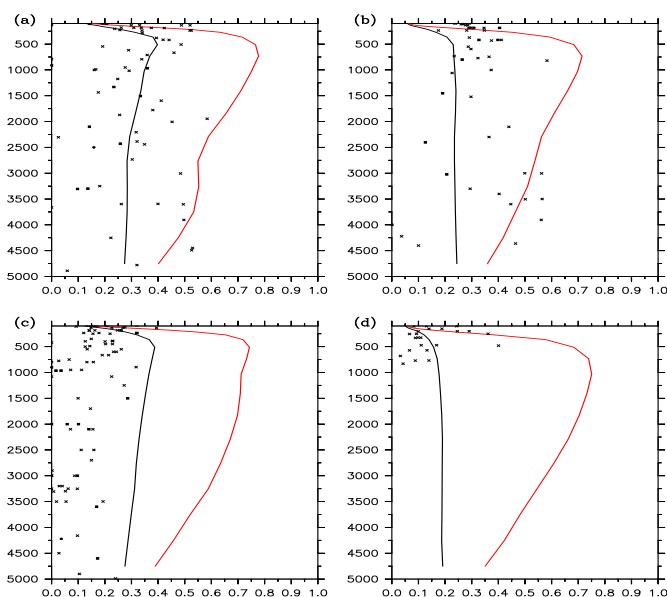

**Figure 5.** The modelled and observed relative contributions of large POC to total POC concentration: bPOC/(sPOC+bPOC) (µM) in different regions of the ocean: **(a)** western, **(b)** oligotrophic and **(c)** eastern North Atlantic Ocean (Lam et al., 2015b); **(d)** Hawaii region (Druffel et al., 1992). Solid lines denote concentrations averaged over the regions marked by the blue rectangles on the map of Figure 2: without (in red) and with the reactive continuum (RC) parameterisation (in black). The black speckles are observations in the respective regions from Druffel et al. (1992); Lam et al. (2015b).





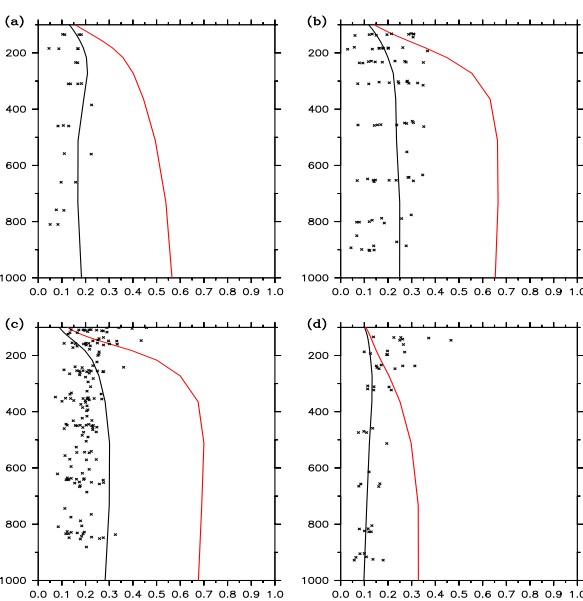

**Figure 6.** Same as Figure 5 for additional regions of the ocean: **(a)** northwest, **(b)** northeast, **(c)** central and **(d)** southern Pacific Ocean. Observations (black speckles) are from Lam et al. (2011).





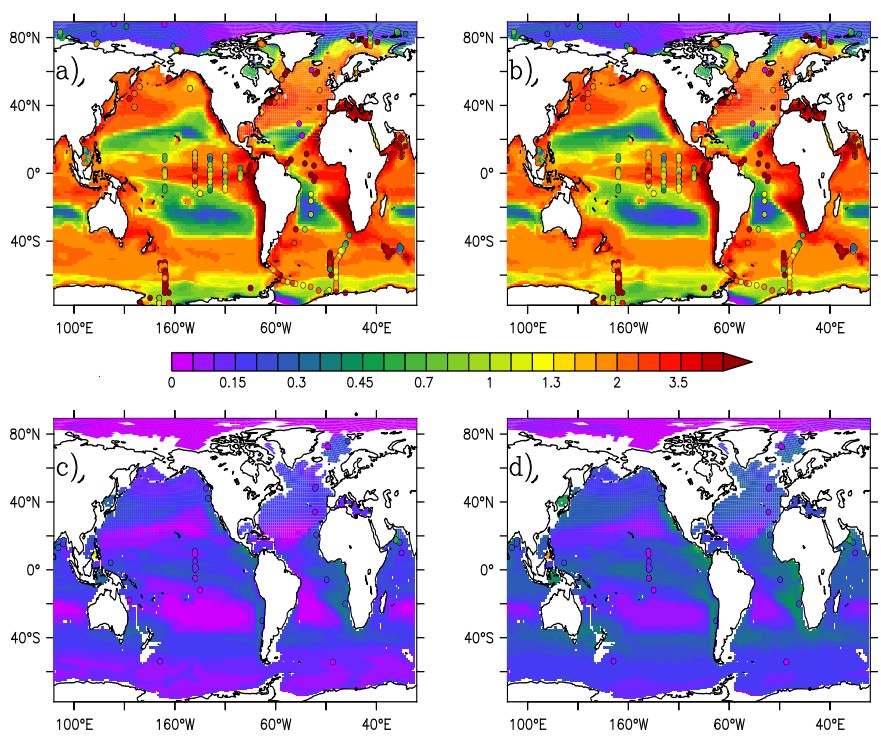

**Figure 7.** Sequestration of POC. The panels on the left show POC fluxes for the simulation without the lability parameter-isation, i.e. POC and GOC have a single lability (`noRC`). Those on the right are with the lability parameterisation (`RC`). The upper two panels (a, b) show the modelled flux at 100 m; the observations are from the same depth ±20 m. Panels (c) and (d) show the flux through 2000 m; the observations are at the same depth ±400 m. The datasets used in these figures are from Dunne et al. (2005), Gehlen et al. (2006) and Le Moigne (2013).





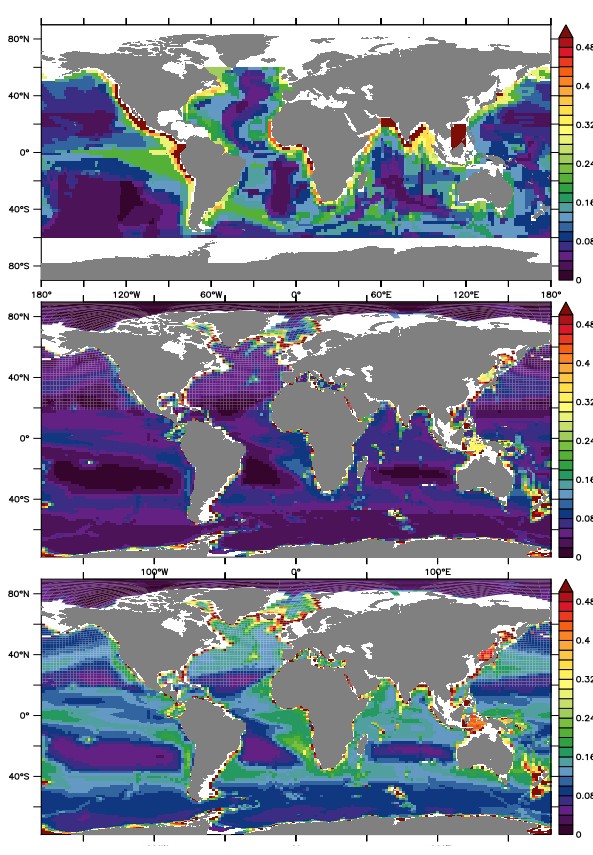

**Figure 8.** POC sedimentation $(\mathrm{mol\,m^{-2}\,yr^{-1}})$ between (a) observations derived from oxygen fluxes (Jahnke, 1996), compared with the model: (b) without lability and (c) with the lability parameterisation.





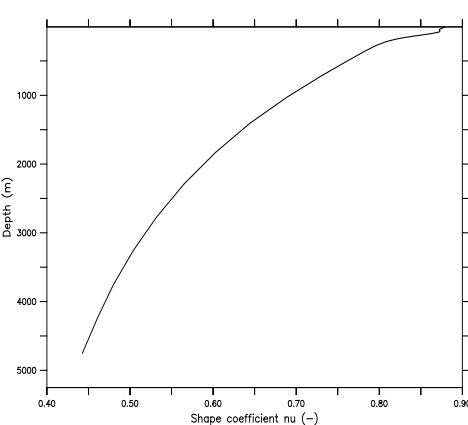

**Figure 9.** Vertical profile of the annual mean shape coefficient $\nu$ (-) of the lability distribution of total POC averaged over the global ocean. $\nu$ has been computed from the results of the RC experiment.





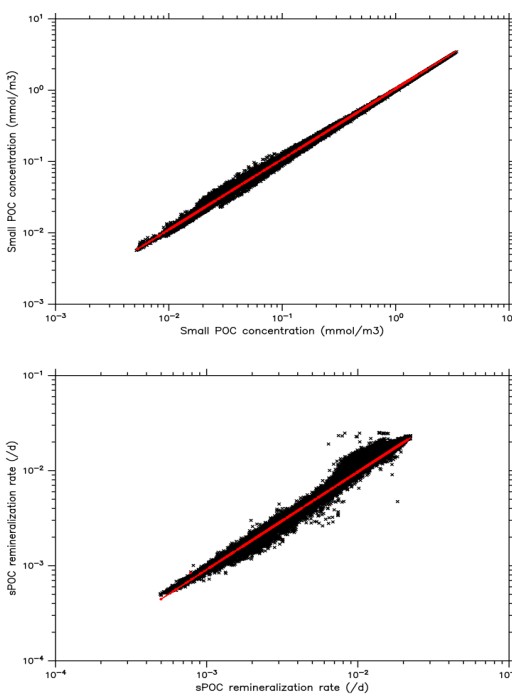

**Figure 10.** Scatter diagrams of the small POC concentration (in $\mathrm{mmol\,m^{-3}}$, top panel) and the remineralisation rate (in $\mathrm{d^{-1}}$, bottom panel). On both panels, the x-axis corresponds to the results from a model experiment in which nine lability classes are explicitly modeled whereas the results from the RC experiment are shown on the y-axis. The red line on both panels shows a linear regression. On the top panel, the slope is 0.99 and the $r^2$ is 0.99. On the bottom panel, the slope is 1.03 and the $r^2$ is 0.98.





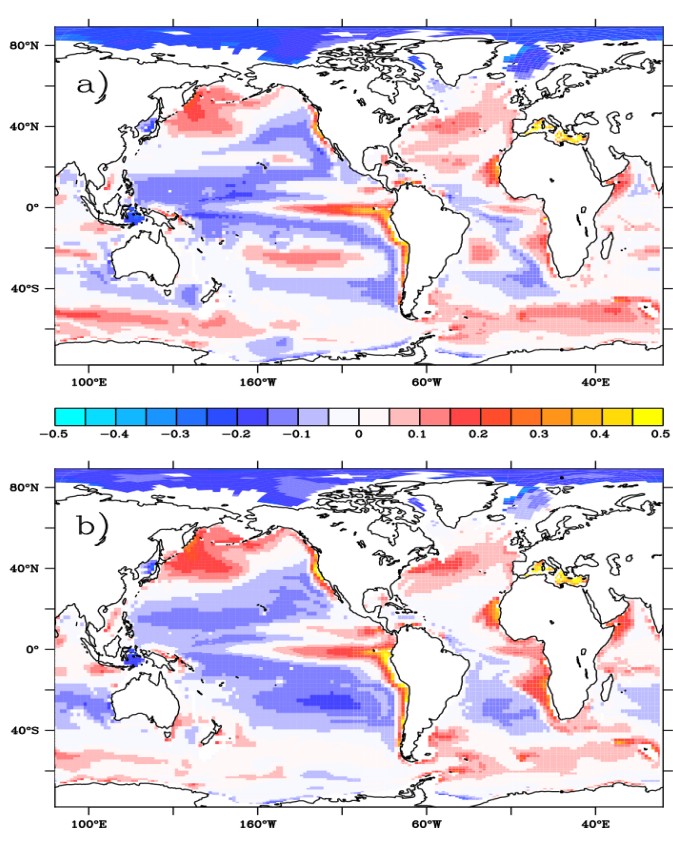

**Figure 11.** Spatial variations of the annual mean anomalies of the remineralisation coefficient $b$ from the global median value:
(a) from the `NoRC` experiment, and (b) from the `RC` experiment. The global median values of $b$ are 0.87 and 0.7, in the `NoRC`
experiment and in the `RC` experiment respectively.




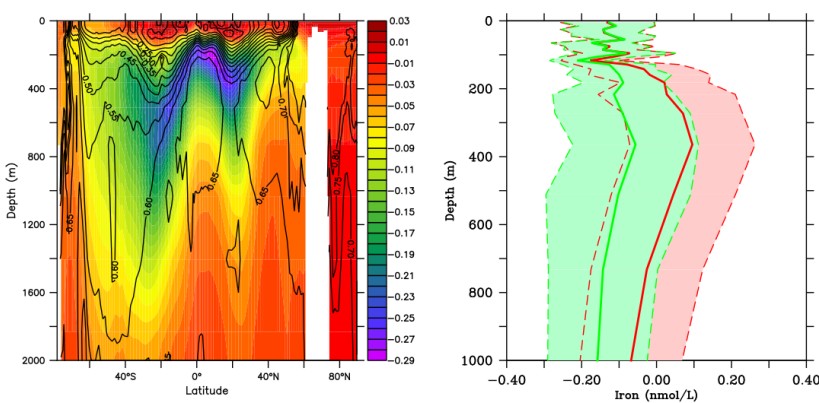

**Figure 12.** Iron distribution (nmol Fe/L) in the Pacific ocean. The left panel shows the zonal average of the difference between the `RC` and the `NoRC` experiments. The isolines display the zonally averaged iron distribution as simulated in the `NoRC` experiment. The right panel shows the vertical difference of the difference the simulated iron distributions (red solid line: the `NoRC` experiment; Green solid line: the `RC` experiment). The light shaded areas display the standard deviation of the difference (red: the `NoRC` experiment; Green: the `RC` experiment).



**Table 1.** Model simulations.

| Experiment | #lability classes | $\lambda^\star_{POC}$ |
|---|---|---|
| noRC | 1 | 0.025 d$^{-1}$ |
| RC | 15 | 0.035 d$^{-1\text{a}}$ |

[a] Remineralization rate of freshly produced POC.

**Table 2.** Statistical model–data comparison of the two simulations for the whole particle dataset displayed in Figure 2. The upper ocean has been excluded from the analysis (defined as less than 200m).

|  | RC | noRC |
|---|---|---|
| $n$ | 2656 | 2656 |
| $r$ | 0.7 | 0.77 |
| RMSE | 0.14 | 0.08 |
| B | 0.12 | 0.02 |
| RI | 4.9 | 1.6 |
| MEF | -0.6 | 0.71 |

**Table 3.** POC production and fluxes global-ocean budget for the different simulations, and estimates based on observations, all in units of PgC/yr of carbon. The numbers in parentheses denote the fluxes due to small POC.

| Simulation | primary production | $\Phi_{\text{photic zone}}$ | $\Phi(z = 2000\,\text{m})$ | $\Phi_{\text{sed}}(z > 1000\,\text{m})$ |
|---|---|---|---|---|
| NoRC | 52 | 9 (3.6) | 0.56 (0.01) | 0.19 (0) |
| RC | 41 | 8.1 (3.6) | 0.81 (0.17) | 0.42 (0.11) |
| Published estimates | 40–60[a] | 4–12[b] | 0.33–0.66[c] | 0.5–0.9[d] |

[a] from Carr et al. (2006);

[b] from Siegel et al. (2014), Laws et al. (2000), Lutz et al. (2007), Dunne et al. (2007) and Henson et al. (2011);

[c] from Henson et al. (2012b) and Guidi et al. (2015);

[d] from Jahnke (1996) and Seiter et al. (2005);