# Peer review of "A reactivity continuum of particulate organic matter in a global ocean biogeochemical model"

_Biogeosciences, 2016_

## Referee Comment (RC1) · J. Dunne (Referee) · 8 Oct 2016

The manuscript "A reactivity continuum of particulate organic matter in a global ocean biogeochemical model" by Aumont et al describe the implementation and implications of a relatively sophisticated representation of sinking particle degradation into a global biogeochemical model. In this representation by Boudreau and Ruddick (1991) the decreasing lability of particles over time is represented as a series of particles with a continuum representation of remineralization rate constant. As this formulation is relatively numerically intensive, its effects have generally been either ignored or approximated simply by an increasing sinking velocity or power-law remineralization rate constant dependence with depth. Thus, the present manuscript makes a powerful contribution to the ocean biogeochemical literature by reconciling all of: past contradictions between particle concentration and flux, particle age distributions, and regional

variability in both empirical power law attenuation coefficients. There are four topics on which I would like a bit more information added to the manuscript: 1) What generates the regional variability in the transfer efficiency in Figure 11? 2) How good is the representation of transfer efficiency as a power in Figure 11? The authors should show some representative profiles of particle flux and the inferred power law fit to go along with the concentration profiles in Figures 3-6. A plot of particle weighted sinking velocity and lability would be most helpful. 3) Is the relationship between initial composition and final remineralization profile amenable to the creation of a numerically efficient metamodel that would avoid the addition of the extra particle tracers? 4) Does this result finally solve the challenge of distinguishing between the two hypotheses of increasing sinking velocity with depth and decreasing lability with depth leading to the fidelity of the Martin curve? The model seems to include both factors, and this reconciliation should be highlighted. Specific comments (For line numbers, I am using the guide on the left rather than counting from the top): P3, ln13 – "explicitly" should be added before "taken into account" since the Martin curve is an implicit implementation of this. P4, ln 3, 10, and p6, ln 6 – "big" should be "large" – also, the nominal size cutoff should be provided. P7, ln 14 – Wilf Gardner has a database (http://people.tamu.edu/~wgardner/~pdgroup//SMP_prj/DataDir/SMP-data.html). Jim Bishop may have one as well. P7, ln 25 – The authors could also consult the Honjo dataset for 2000m values (http://usjgofs.whoi.edu/mzweb/smppi/honjo.html) P8, ln 18 – "does represent" should be "represents", and the two "the" should be removed. P8, ln 21 – What are the units of "0.1 to 0.4"? P8, ln 24 – Remove "associates to a relatively strong remineralization" P9, ln 29 – What are these modeled and observed C14 ages in confict? P9, ln 15 – add "a" before "result" P9, ln 16 – Add "variable" before "lability" P9, ln 19-20 – To make the case the the model is good, one compares with observation, but if both models are not good, the improvement in r2 relative to observations will not necessary be high. . . to show the result of the parameterization has changed the model distributions, one should show the r2 for both mod-obs and mod-mod. P9, ln 23 – What is the third simulation "in all three simulations" P10, ln 1-10 – The authors need

not be concerned that their estimates to not agree with the anomalously low Henson et al. (2012) and follow on Guidi et al., (2015) relative to the previous literature by Yamanaka and Tajika (1996), Schlitzer (2000), Laws (2000), Muller-Karger et al (2005) and Dunne et al (2007). While more recent, these "revised" Henson estimates are deeply flawed and nicely refuted empirically in Weber et al., (2016; PNAS). The problem with the Henson et al. (2012) and follow on Guidi et al., (2015) papers is that they restrict their calibration to particle export estimates that have the advantage of being geographically broad through the tropics but were found to suffer from strong analytical biases. This was first identified by Quay (1997) who performed a carbon budget for the equator at 140 as part of the EqPac synthesis and demonstrated that the Buesseler method of measuring C:234Th ratios via in-situ filtration led to inferred carbon fluxes a factor of two lower than necessary to achieve a steady state carbon budget for the region. It turned out that the filters were directly absorbing dissolved 234Th, leading to artificially low C:234Th ratios and inferred carbon export fluxes. Unfortunately, this analytical bias was not fully appreciated until the field program was complete and an extensive database was developed. While Buesseler soon discontinued use of this method and critically championed the development of neutrally buoyant sediment traps to achieve more accurate estimates, these filter-based C:234Th ratio particle export estimates remain in the JGOFS-era database. As a result, scientists continue to publish syntheses based on these data. If these are to be used, they should only used in combination with the suite of other estimates that are all double the global flux with much more tropical contribution. P11, ln 30 – "relied on" should be "tests" P12, ln 28 – What are the small particle ages in these simulations? P12, ln 5 – Again, don't just trust the Henson numbers. P13, ln 2 – Some discussion of the modeled ecological factors driving the regional varaibilty in transfer efficiency is warranted. P15, ln 10-14 – Provide the r2 or RMSE comparison for these runs. P16, ln 10 – It would be extremely helpful to the ocean biogeochemical modeling community to have a parameterization of the transfer efficiency based on the ecological structure to avoid the addition of the extra particle tracers. P17, ln 24 – A note on the implications for variability in sinking

particle weighted lability and velocities as a function of depth Figure 4 – This should be combined with Figure 3 into a single figure. Figure 6 – This should be combined with Figure 5 into a single figure. Figure 11 – Missing colorbar

---

## Referee Comment (RC2) · Anonymous Referee #2 · 4 Nov 2016

The paper introduces a new scheme to parameterize a spectrum of particle lability classes in in global biogeochemical model of the marine pelagial. The scheme follows approaches that have been developed for benthic degradation (Boudreau and Ruddick, 1991), and applies these in modified form to the pelagic realm.

I think the approach is a very interesting and new one. However, I have several concerns about the way it is presented, which hampers understanding and evaluation of the approach. My concerns are related to the method description and comparison to (and distinction from) other models, that include different detritus classes, or otherwise attempt to represent the particle continuum with depth and time. These are explained in detail below:

Method:

I had some difficulties to understand how exactly lability is parameterized in this model; what I understood from the method description is the following:

After an overview about the different types of lability models (multi-G, RC), the authors present and discuss the RC model applying a Gamma function, namely g(k,0)=gamma(nu,ak)/Gamma(nu) (Eqn. (1)). On page 6, line 7-8 they state that "for the sake of simplicity, we assumed that the shape factor nu is equal to 1."

In case nu=1, Eqn. (2) becomes a simple exponential function exp(-ak). Further derivation yields G(0)=g_0/a (integrating over k from 0 to infinity), and dG/dt = G/(a+t), as in their Eq. (4). This derivation was also carried out by Boudreau and Ruddick (1991); they even went a step further, derived and described the decay of organic matter as a nu'th order of organic matter concentration: dG/dt=-nu/(a*G(0))*G^(1+1/nu) (their eqns 41 and 42). Thus, nu=1 results in a second order process, with dG/dt=1/(a*G(0))*G^2, which would be easy to implement in any model, the restriction being the assumption of a closed system, and 0 =< k =< infinity.

However, on page 5, lines 17ff the authors rightly note that this continuous form is difficult to apply in a global model, because of the many processes that can distort the surface distribution on its way through space and time. Therefore, they seem to discretize the lability spectrum in 15 classes for each POC component, resulting in a multi-G model of particle lability, where the boundaries of the lability spectrum are defined by -ak_(i+1) and -ak_i, and the fraction each class occupies in the reactivity continuum is given by Eqn. (6), namely G(bar(k)_i,0)= exp(-ak_(i+1))-exp(-ak_i). (All assuming nu=1, i.e. Gamma(1) = 1).

Then, on p. 6, line 12 the authors "that the lability distribution of POC is insensitive to ocean transport and is only modified by sinking, by the biological sources and sinks, and by vertical mixing in the mixed layer.", thereby reducing the model to a 1D system. Based on this assumption, they then exchange the time axis by z, via t=z/w, and solve the system analytically. For nu=1, the outcome will then be:

G(k,t) = g_0 * a/(a+f(T)*z/w)*(exp(-(a+f(T)z/w)k_(i+1))-exp(-(a+f(T)z/w)k_(i)))

Provided I understood everything correctly, my questions are now:

(1) What exactly is the difference between this model, and a model that uses several (15) discrete lability classes at the surface, whose remineralization is distributed instantaneously over depth according to their constant remineralization rate and sinking speed? Of course, by choosing the Gamma function for the distribution over the lability space, and hence their average lability, it avoids the necessity to select for many different parameters (this choice to went to assigning "a" and nu). It is difficult to determine differences this from Eqn. (7) alone, as neither sources nor sinks are specified; it also seems that the indices for the different lability classes are missing. I suggest to extend on this part, and shorten the one about the gamma function (see 2), or (this relates to the nu=0.16 experiments) comment on the way the gamma function was computed (if necessary).

(2) Why not start explaining the method before the background of nu=1, and refer to Boudreau and Ruddick (1991) for a complete (and in my opinion very thorough) review of the merits and pitfalls of the incomplete gamma function to represent the lability spectrum of organic matter? As you rightly note, math becomes much easier with nu=1, and the paper will be more susceptible for a greater audience. On the other hand, I would suggest to be much more specific about the coupling of the lability spectrum to biogeochemistry: many parameters and variables are not defined and explained (a, g_0, S, P). Decay is denoted both in terms of k_i, bar(k)_i, f(T), bar(lambda) ... This is very confusing, particularly as the model is all about decay.

(3) How do you arrive at Eqn. (4)? It seems to result from a Laplace transform; if so, I would mention it or refer to some source (again, Boudreau and Ruddick seem to be a good source). This also applies to eqns. (9) and (10).

(4) It seems that one of the assumptions if the discretized model is that particles within a lability class do not change their properties over time (or space). If so, I would mention

it briefly.

(5) The analytic integration over depth requires that w=const., correct? I suggest to state this briefly; this is in contrast to the common notion that particles become faster with depth, and might help to distinguish this model from other models making that assumption.

Results:

The good fit to observed POC profiles (Fig. 3-6) is really impressive at first sight. To make it even more convincing, you could explain what exactly sPOC and bPOC is (Fig. 5, 6). I also wonder whether such a good fit could be obtained using the standard 2-component PISCES model, with different/tunes decay rates for the small compartment. Did you ever test this?

Figure 7, lower panels: please choose a different colour scale. It it very difficult to see the dots for observations. (At first sight I thought there were no observations at all.) Further, a plot such as Fig. 3 of Gehlen et al. (2006) would be very nice - I know that comparison to trap data always shows a huge scatter, but it might nevertheless help the reader form an opinion about model performance ("Has anything improved?")

Figure 8 looks very impressive. Did you also compare against the data set by Seiter et al.? As far as I know, the data are available, and a comparison against this data set possibly includes less assumptions (stoichiometry; interpolation).

Figure 9 and p. 12, lines 13ff: How did you compute nu from the RC model? As the model's assumption was nu=1, where do these variations come from? Beside variation in nu, Boudreau and Ruddick found a huge range of "a"'s, from months to thousands of years. If applicable, did you try the same approach for "a" (perhaps better: apparent nu and apparent "a", in contrast to the prescribed "a" and nu)?

Discussion:

The discussion repeats some things that were already presented in the method description. Then, several other experiments (nu=0.16, p. 12; 9 lability classes, modeled both implicitly and explicitly ) and model - data comparisons ("apparent" nu, Fig 9 and p. 12; iron distribution, p. 15 and Fig.) are are presented or mentioned briefly. I suggest to move these to the results section, and also explain their setup in the methods section. How was the nu=0.16 experiment computed? Did it require numerical evaluation of the incomplete gamma function, and if so, how expensive was these in terms of computation time?

Given the quite large uncertainties associated with the parameterization of the iron cycle (Tabgliabue et al., 2015; 10.1002/2015GB005289) I find a comparison against dissolved iron concentrations quite ambitious; nevertheless, I could not see any observations of iron in Fig. 12 (Does "vertical difference of the difference the simulated iron distribution" mean "difference between model and observations"?) Do simulated nutrients or oxygen differ in the same way as iron does?

Finally, there have been two recent papers that attempt to simulate the degradation of particles, in local or global models, namely Yokulsdottir and Archer (2016, www.geosci-model-dev.net/9/1455/2016/) and deVries et al. (2014, www.biogeosciences.net/11/5381/2014/) - perhaps some discussion about the assumption of this model in contrast to these (and possibly others) might encourage the reader to dive deeper into this very interesting topic.

————————————————————————

Specific comments:

- p. 1, line 13: The particle flux profile introduced by Suess is not an exponential one, but is described by 1/(a*z+b)

- p.6, Eqn (7) What are the P-term and S-term exactly?

- p.6., line 20 "Assuming constant sources and sinks over each grid cell" - constant over what? Time?

- Table 1: The number of lability classes refers to each POC compartment, correct? Could you give the value for "a" for experiment RC? Possibly also the range of k_i (i.e. the lowest and highest value)?

- Table 2: It seems to me as if the headers (RC, noRC) are swapped. Please comment on the different acronyms for the metric: instead of "B" one could also write "bias". It is quite inconvenient to search through the paper for RI and MEF.

- Table 3: What exactly is POC production - production of fecal pellets? Primary production (as in the table header)? What is Phi? I assume sedimentation through a depth horizon of at the sea floor - but this should be clearly stated in the caption.

---

## Author Comment (AC1) · 29 Nov 2016

**Response to Reviewer 1: John Dunne**

We would like to thank Dr. John Dunne for his very positive and interesting comments on our manuscript. In the following response, we first address the four general concerns and comments of his review. In a second part, we answer to his more specific comments.

**What generates the regional variability in the transfer efficiency in Figure 11?**

In Figure 11, we display the transfer efficiency of the export through the mesopelagic domain. In fact, there are several manners to diagnose the transfer efficiency. For instance, in Henson et al. (2012), two different definitions have been used: 1) the ratio of the export at 2000m over the export at 100m, i.e. $T_{eff} =^{2000} F/^{100}F$, and 2) the value of the Martin's b coefficient. In our study, we have chosen the second definition. On Figure 11, we compare the annual mean anomalies of this Martin's b coefficient from the global median value. As discussed in the manuscript, part of the regional variations, i.e. the differences between the two experiments used in our study, is explained by the lability parameterization. However, in the standard experiment, which does not include the new lability parameterization, the **b** coefficient still exhibits very significant spatial variations.

These variations stem from three different dominant processes. First, zooplankton grazes upon particulate organic matter. In PISCES-v2 (the version of PISCES that is being used here), two types of grazing by zooplankton are represented. In addition to the conventional concentration-dependent grazing, POC is also consumed by flux-feeding (see Equation 29 in Aumont et al. (2015)). According to our model, flux-feeding dominates by far in the mesopelagic domain. To infer the impact of flux-feeding on the regional distribution of the transfer efficiency, we have performed an additional sensitivity experiment in which this process has been removed. Figure 1 shows the transfer efficiency computed from this experiment together with the transfer efficiency in the standard experiment. First, the global median value differs quite substantially from the standard experiment, i.e. 0.6 instead of 0.87. Second, the regional distribution is strongly altered. When flux-feeding is omitted, the $b$ coefficient tends to be high in the subtropical gyres, close to the global median value in the productive areas of the low latitudes and very low in the high latitudes, especially in the Southern Ocean. Thus, flux-feeding plays a very important in shaping the regional patterns of the transfer efficiency. In particular, it tends to strongly reduce the transfer efficiency in the very productive zones of the low latitudes.

The second process which explains the regional variations of the Martin's $b$ coefficient is the relative contribution of the big particles to total POC in the upper ocean. A large contribution of these big particles tends to lead to a high transfer efficiency because their sinking speed is large and thus, their remineralization length scale is long. Conversely, a pool of POC dominated by small particles will tend to generate low transfer efficiency. This process explains the large values of the Martin's $b$ coefficient in the sensitivity experiment presented in this response (see panel a) in Figure 1 of this answer). Finally, the third dominant process is temperature. The remineralization rate of POC is made a function of temperature in PISCES (see section 4.1.1 in Aumont et al. (2015)). Marsay et al. (2015) have proposed in their study a detailed analysis of the impact of temperature on the export of POC.

In the submitted version of the manuscript, these three processes are listed and discussed briefly in section 4.2 on page 14. We could provide a much more detailed analysis and add some figures such as Figure 1 of this review. However, since the

[Figure]

Figure 1: Spatial variations of the annual mean anomalies of the remineralisation coefficient $b$ from the global median value: (a) from the new sensitivity experiment with no flux feeding, and (b) from the `NoRC` experiment. The global median values of $b$ are 0.87 and 0.6, in the `NoRC` experiment and in the sensitivity experiment respectively.

primary focus of this paper is the impact of a variable lability on the distribution of POC and on the export of carbon, we believe it would distract the readers from this main focus.

**How good is the representation of transfer efficiency as a power in Figure 11? The authors should show some representative profiles of particles ...** Our main purpose on Figure 11 was to discuss the spatial patterns of the transfer efficiency in the mesopelagic domain and the impacts of lability on this transfer efficiency. As mentioned above, there are several manners to diagnose the transfer efficiency. We have chosen a diagnostic based on the Martin's $b$ coefficient because of the widespread use of Martin's parameterization in ocean biogeochemical models. However, this does not mean that our predicted fluxes follow a power law function. Following the reviewer's suggestion, we have investigated in the `RC` experiment how close the predicted fluxes are to a power law distribution. To do so,

we have compared the predicted fluxes to reconstructed fluxes using the $b$ coefficient diagnosed on Figure 11.

Figure 2 displays several statistical indicators of this comparison. The correlation coefficient is close to 1 over large regions of the ocean, especially in the low latitudes. In the high latitudes, it is lower. This would suggest that a power law function could be a good approximation of the simulated fluxes over large areas of the global ocean. However, the correlation coefficient in that case is not necessarily a good indicator of the fit as suggested by the other two indicators. First, the slope often diverges significantly from 1. Second, the normalized RMSE can be very large (above 2), especially in productive areas. In fact, the high value of the correlation coefficient stems from the general vertical shape of the vertical fluxes which decrease sharply with depth in a very convex manner.

As a consequence of that analysis, we think that a power law function is not a satisfactory approximation of the predicted fluxes in our study. Thus, the Martin's b coefficient displayed on Figure 11 should be interpreted as a diagnostic of the transfer efficiency in the mesopelagic domain, not as an attempt to describe the fluxes with a power law function. In the revised version of the manuscript, we added on page 13 a discussion on the use of the Martin's b coefficient: *"Figure 11 displays the anomalies of b relative to the median value of that coefficient, both in the `NoRC` and in the `RC` experiments. The b coefficient is used here as a diagnostic of the transfer efficiency of POC in the mesopelagic domain. In fact, a close inspection of the vertical profiles of the simulated vertical fluxes of POC shows that they can diverge significantly from a power law distribution, especially in the high latitudes and in very productive areas (see Figure S1, in the supplementary materials)."*

**Is the relationship between initial composition and final remineralization profile amenable to the creation of a numerically efficient metamodel ...** Computing cost is always an issue in global ocean biogeochemical models. New parameterizations often imply a substantial extra cost and any means to overcome this extra cost is beneficial. Furthermore, metamodels can be powerful and efficient tools to reconstruct fluxes from incomplete data, such as satellite data for instance. Thus, we agree with John Dunne that deriving a metamodel from our model experiments would be of a great value.

In this study, we have not attempted to derive such a metamodel as our primary objectives were 1) to investigate the impacts of a variable lability of POC on the distribution and vertical fluxes of POC, and 2) to attempt to reconcile realistic fluxes and POC concentrations. Nevertheless, is the construction of a metamodel feasible in that specific case? Unfortunately, we cannot give a certain answer to that question. As discussed in the manuscript and in the first item of this response, the vertical structure of the fluxes depends on the size structure of POC in the upper ocean (the relative contribution of big particles), the abundance and the vertical distribution of zooplankton and the vertical structure of temperature. The two latter points rely on 3-D fields which do not necessarily correlate well with upper ocean variables. For instance, the vertical structure of zooplankton is impacted in a non linear (and thus non simple) way by oxygen but also by the vertical structure of the concentrations and the fluxes of POC. This should make the construction of a metamodel quite challenging. Furthermore, as discussed in the second item and as displayed on Figure 2 of this response, the vertical fluxes of POC can significantly deviate from a simple power law function over large regions of the ocean. As a

[Figure]

Figure 2: Statistical comparison between the simulated vertical fluxes of POC in the `RC` experiment and reconstructed fluxes using a power law distribution with the $b$ coefficient displayed in Figure 11. Panel (a) shows the spatial patters of the correlation coefficient $r^2$. Panel (b) shows the slope of the linear regression analysis. Panel (c) displays the normalized RMSE between both fluxes.

consequence, our feeling is that the construction of a robust metamodel should be difficult. A substantial additional analysis would be necessary which exceeds the primary objectives of our study.

Setting aside the construction of a metamodel, a related point is whether a more numerically efficient model can be constructed from our model. In our study, numerical efficiency has been an important issue which explains the quite strong assumption we made by neglecting the impact of advection and diffusion. This assumption avoids the explicit representation of the computationally intensive lability spectrum. As a consequence, the extra-cost of the lability parameterization is limited to about 20%. Despite being reasonable, this extra-cost is still significant. To considerably reduce this cost, one can be tempted to make the assumption of a closed system for POC. In that case, the model simplifies to Equation 4 of the manuscript. Figure 3 compares the vertical lability distribution of small POC using that strong assumption with the prediction based on the RC model. Differences are very large and can reach almost one order of magnitude in the interior of the ocean. Thus, the assumption of a closed system is not valid. The sources and sinks of POC in the interior of the ocean play a major role on the vertical and horizontal structure of the lability distribution.

In the revised version of the manuscript, we extend the discussion on the computation cost of our parameterization at the end of section 2.2 and add Figure 3 of this comment as Figure 2 in the revised version of the manuscript: *"The lability parameterisation introduces an extra cost of about 20%, but it depends of course on the number of lability pools. To further considerably reduce this extra-cost, one could be tempted to adopt the assumption of a closed system. In that case, the model simplifies to Equation 4 for both small and large POC. Figure 2 compares the vertical lability distribution of small POC using that strong assumption to the prediction using the complete lability parameterization. Differences are large and can reach almost an order of magnitude in the interior of the ocean. Thus, the assumption of a closed system introduces large errors. The sources and sinks of POC in the interior of the ocean play a major role on the vertical and horizontal patterns of the lability distribution."*

**Does this result finally solve the challenge of distinguishing between the two hypotheses of increasing sinking velocity with depth and decreasing lability with depth?** Unfortunately, the answer to that question is no. In our model, vertical sinking velocities are constant and lability is decreasing with depth. Nevertheless, our feeling is that the most probable hypothesis is a decreasing lability with depth. As a clue to support that hypothesis (it is a clue, not a demonstration), the assumption of an increasing sinking velocity with depth implies that the vertical variations of POC could be described as a power law function with an exponent equal to $-(b+1)$. So POC concentrations should decrease quite strongly with depth in the deep ocean which is not supported by the observations (see Figure 3 of the manuscript for instance). An alternative would be that only the fraction of POC that contributes the most to the vertical fluxes, i.e. the big particles, decreases with depth. In that case, the relative contribution of the big particles should be decreasing with depth. Again, this is not supported by the observations (see Figures 5-6). As already stated, this does not demonstrate that mean sinking velocities do not increase with depth, but simply suggests that such is not case.

In the rest of this response, we address the more specific comments made by the

[Figure]

Figure 3: Vertical distribution along the equator of the ratio between the remineralization rate of small POC computed when the assumption of a closed system is made and the remineralization rate computed in the standard `RC` experiment.

reviewer.

**P3, ln 13 - "explicitly" should be added before ...** Done.

**P4, ln 3, 10, and P6, ln 6 - "big" should be "large"** This has been changed. The nominal size cutoff is 100 $\mu$m. This indication has been added to the model description on page 4.

**P7, ln 14 - Will Gardner has a database . Jim Bishop may have one as well.** We thank John Dunne for this information. We were not aware of that database. Following John Dunne's advice, we have downloaded the database from the given website. In fact, this dataset is not based on POC observations but include beam attenuation observations. Then a regression relationship should be applied to derive POC concentrations from beam attenuation values. After inspecting the related publications (i.e., Gardner et al., 2006), the first issue we faced is that the relationship has been built mainly from surface data. For small values, typically measured in the interior of the ocean, the scatter is extremely large (see for instance Figure 3 in Gardner et al. (2006)) which makes the relationship not very robust, at least for the mesopelagic and deep domains. The second related issue comes from the absolute value given by the relationship. There is a minimum value given by the intercept of the regression between beam attenuation and POC and at least, for the Pacific ocean, this intercept is very high (larger than 2 $\mu M$), almost an order of magnitude larger than typical directly observed values. Figure 4 compares the mean vertical profiles of POC over the Pacific and Atlantic Oceans as derived from Gardner's database to the profiles computed from the database shown on Figure 2 of our manuscript. They differ a lot which makes the use of both challenging.

[Figure]

Figure 4: Vertical profiles of POC ($\mu$M) averaged over the Pacific (black) and the Atlantic (red) oceans. Solid lines correspond to the data presented on Figure 2 of the manuscript. Dashed lines display the POC data reconstructed from Gardner's dataset.

Furthermore, POC concentrations in the deep ocean reconstructed from the beam attenuation observations exceed in the Pacific Ocean those in the Atlantic ocean by almost an order of magnitude which is hard to explain by biogeochemical arguments. For those reasons, we prefer not to use Gardner's database in our study. We have not been able to find any publicly available database from Bishop's group.

**P7, ln 25 - The authors can also consult the Honjo dataset for 2000m values** Again, we would like to thank John Dunne for his suggestion. In fact, most of the data in Honjo dataset were also included in the dataset from Gehlen et al. (2006). However, some were missing. Figure 5 has been redone to include those missing data.

**P8, ln 18 - "does represent" should "represents" ...** This has been changed.

**P8, ln 21 - What are the units of "0.1 to 0.4"?** There are no units since this is a relative contribution.

**P8, ln 24 - Remove "associates to a relatively strong remineralization"** We don't think we should remove those words because the sinking speed by itself is not enough to explain the sharp decrease in POC with depth.

**P9, ln 29 - What are these modeled and observed C14 ages in conflict?** We do not model C14 in POC here. The observed C14 isotopic ratio in suspended POC suggests that this pool is quite older than sinking POC. This observation is in contradiction with the prediction of the standard version of the model which simulates a slow-sinking POC pool that is very young. We have changed this line to make our point clearer.

**P9, ln 15 - add "a" before "result"** Done.

**P9, ln 16 - add "variable" before "lability"** Done.

**P19, ln 19-20 - to make the case the model is good, one compares with observations ... one should show the r2 for both mods-obs and mods-mods** We agree with John Dunne that a comparison based on the value of $r$ is not sufficient to assess the performance of a parameterisation. In particular, this does not tell if two different parameterizations produce significantly different results, especially if we restrict the computation to a model-data comparison. To overcome that limitation, we computed in the submitted version of the manuscript other statistical indicators relative to the observations. Since the performance of the two model configurations is very different, especially the scores based on the MEF and RI indices, the POC distributions should significantly differ between the two experiments. Following John Dunne's suggestion, we have also computed the correlation coefficient between the `RC` and `NoRC` experiments which is equal to 0.98. Thus, it is very high which would suggest that both model configurations produce very similar results in terms of POC distribution. However, in that specific case, the correlation coefficient is not necessarily the best index (see our response to general comment 1). When we compute the RI index, its value over the global ocean is 14.6 which indicates on the contrary that both models strongly differ. We did not change the manuscript because we think that the different statistical indices provided in our study are sufficient to prove that the two different models produce POC distributions that are significantly different, following the recommendations given by Stow et al. (2009) and Doney et al. (2009).

**P9, ln23 - What is the third simulation "in all three simulations"** This is a mistake. There are only two simulations. Three has been changed to two.

**P10, ln 1-10 - The authors need not be concerned ...** We deeply thank John Dunne for his detailed analysis of the estimates provided by Henson et al. (2012) and Guidi et al. (2015). We were not aware of the recent study by Weber et al. (2016). This study clearly challenges the transfer efficiencies found in Henson et al. (2012) and Guidi et al. (2015) and suggests larger export to the deep ocean. The horizontal patterns of this transfer efficiency seem also in qualitative agreement with what Marsay et al. (2015) found. We will change the text in section 3.2 to include a discussion on the potential biases in the estimates of Henson et al. (2012) and Guidi et al. (2015).

**P11, ln 30 - "relies on" should be "tests"** Changed.

**P12, ln 28 - what is the small particle ages in these simulations?** We don't really understand that question as the ages of the particles are not mentioned here. Nevertheless, the ages of the small particles in the `NoRC` experiment never exceeds a few weeks. In the `RC` experiment, the ages of these small particles exceed 5 years at the bottom of the ocean.

**P12, ln 5 - Again, don't just trust the Henson's numbers** The numbers of the $b$ coefficients found in the studies by Henson et al. (2012) and Guidi et al. (2015) are only used to illustrate that this coefficient should not be considered constant.

**P13, ln 2 - Some discussion of the modeled ecological factors driving the regional variability in transfer efficiency is warranted** Please see our detailed answer to the first general comment.

**P15, ln 10-14 - Provide the r2 or RMSE comparison for these runs** Following John Dunne's suggestion, we have computed the RMSE of the nitrate and oxygen distributions between the two model experiments. They are equal to 0.85 and 17.4 $\mu$M respectively. The text has been changed accordingly to quote these values.

**P16, ln 10 - It would be extremely helpful for the ocean biogeochemical modeling community ...** Please see our response to the third general comments made by the reviewer.

**Figure 4 - This should be combined with Figure 3 into a single figure** Done.

**Figure 6 - This should be combined with Figure 5 into a single figure** Done.

**Figure 11 - Missing color bar** In the submitted version of the manuscript, the colorbar does not seem to be missing on figure 11.

**References**

Aumont, O., Ethé, C., Tagliabue, A., Bopp, L., and Gehlen, M.: PISCES-v2: an ocean biogeochemical model for carbon and ecosystem studies, Geosci. Model Dev., 8, 2465–2513, doi:10.5194/gmd-8-2465-2015, 2015.

Doney, S. C., Fabry, V. J., Feely, R. A., and Kleypas, J. A.: Ocean Acidification, Annu. Rev. Mar. Sci., pp. pp. 169–92, doi:10.1146/annurev.marine.010908.163834, 2009.

Gardner, W. D., Mishonov, A. V., and Richardson, M. J.: Global POC Concentrations from in-Situ and Satellite Data, Deep Sea Research Part II: Topical Studies in Oceanography, 53, 718–740, doi:10.1016/j.dsr2.2006.01.029, 2006.

Gehlen, M., Bopp, L., Emprin, N., Aumont, O., Heinze, C., and Ragueneau, O.: Reconciling surface ocean productivity, export fluxes and sediment composition in a global biogeochemical ocean model, Biogeosci., 3, 521–537, doi:10.5194/bg-3-521-2006, 2006.

Guidi, L., Legendre, L., Reygondeau, G., Uitz, J., Stemmann, L., and Henson, S. A.: A new look at ocean carbon remineralization for estimating deepwater sequestration, Global Biogeochem. Cy., 29, 1044–1059, doi:10.1002/2014GB005063, 2014GB005063, 2015.

Henson, S. A., Sanders, R., and Madsen, E.: Global patterns in efficiency of particulate organic carbon export and transfer to the deep ocean, Global Biogeochemical Cycles, 26, GB1028, doi:10.1029/2011GB004099, 2012.

Marsay, C. M., Sanders, R. J., Henson, S. A., Pabortsava, K., Achterberg, E. P., and Lampitt, R. S.: Attenuation of sinking particulate organic carbon flux through the mesopelagic ocean, Proceedings of the National Academy of Sciences, 112, 1089–1094, doi:10.1073/pnas.1415311112, 2015.

Stow, C., Jolliff, J., Jr., D. M., Doney, S., Allen, J., Friedrichs, M., Rose, K., and Wallhead, P.: Skill assessment for coupled biological/physical models of marine systems, J. Mar. Syst., 76, 4–15, doi:10.1016/j.jmarsys.2008.03.011, 2009.

Weber, T., Cram, J. A., Leung, S. W., DeVries, T., and Deutsch, C.: Deep Ocean Nutrients Imply Large Latitudinal Variation in Particle Transfer Efficiency, Proceedings of the National Academy of Sciences, 113, 8606–8611, doi:10.1073/pnas.1604414113, 2016.

---

## Author Comment (AC2) · 2 Dec 2016

**Response to Reviewer 2**

We would like to thank the reviewer for the interesting discussion on our manuscript. In the following response, we have carefully studied the comments and have made corrections which we hope to meet with approval.

The reviewer starts by a detailed summary of the method we have used to describe the lability distribution of particulate organic carbon. One of the concerns of the reviewer is whether he understood correctly the methodology of the manuscript. Based on the detailed summary he provided in his review, we confirm that he correctly understood our methodology.

**Method:**

1) **What exactly is the difference between this model and a model that uses several (15) discrete lability classes** We are not really sure to fully understand what the reviewer suggests in this comment. Our parameterization is, as mentioned by the reviewer, based on a discretization of the lability space using 15 classes. The number of classes is not a strict constraint and can be freely specified when the experiment is set up. The remineralization rate of each class is constant over the water column by definition. It is also constant over the entire ocean. Sinking speeds are also constant. Thus, in theory, the vertical distribution of each class could be computed easily from the surface distribution. Concentrations would decrease as an exponential function of depth with a length scale defined as the ratio of the sinking speed over the remineralization rate. However, particles are continuously consumed and produced over the entire water column. In addition, particles are exchanged between the small and large pools which have different lability distributions. Thus, predicting the vertical profiles of each lability pool is not feasible. In the revised version of the manuscript, a new figure will be added which compares remineralization rates predicted when a closed system is assumed (basically when the distribution of each lability class can be analytically determined from surface values) to remineralisation rates predicted by the standard `RC` experiment. This figure is displayed as Figure 3 in our response to John Dunne's review. Following the reviewer's suggestion, we have extended the description of the parameterization in section 2.2 of the manuscript. We also better describe the different terms in Equation 7. Indices are also now explicitly written in that equation. The gamma function is computed using the algorithm of MacLeod (1989).

2) **Why not start explaining the method before the background of nu = 1, and refer to ...** We have removed from the model description the assumption that $\nu$ is equal to 1. In fact, this assumption was confusing since the description and the equations of the submitted version of the manuscript were not relying on that assumption. The parameterization we developed for this study can use any value of $\nu$. As suggested by the reviewer, rewriting the equations with the assumption $\nu = 1$ would make the maths much easier but it would also make the parameterization less general. Furthermore, in the discussion section, we mention an experiment in which $\nu$ has been set to 0.16 which would be difficult to explain if our parameterization were based on the assumption $\nu = 1$. As suggested by the reviewer, we have carefully defined all the notations in section 2.2 of the manuscript.

3) **How do you arrive at Eqn. (4)? It seems to result from a Laplace transform; ...** No, it does not result from a Laplace transform. It stems from an integration by

parts of the degradation equation:

$$\frac{dg(k,t)}{dt} = -kg(k,t) \tag{1}$$

The explanation of the integration can be found in Boudreau and Ruddick (1991) and Boudreau et al. (2008). This is now mentionned in the revised version of the manuscript.

**4) It seems than one of the assumptions if the discretized model is that particles within a lability class do not change their properties over time (or space) ...** We are not sure to fully understand what the reviewer means here. In our model, particles are a mixture of compounds with different labilities. This mixture evolves with time in response to different processes such as degradation, grazing, coagulation, ... Thus, the composition of the particles evolves with time and is not constant in time and space. In our model, the lability spectrum corresponds to the compounds which make up particles. In other words, particles are not assigned a constant lability but are made up of compounds which have each a constant lability and whose proportions in particles are changing with time and space.

**5) The analytic integration over depth requires that w=const., correct ? ...** In fact, a constant sinking speed is not required by our parameterization. Vertical profiles of the lability distribution is obtained by analytically integrating Equation 7 over each individual grid cell. This requires that properties such as sinking speed, production, consumption of particles are spatially constant inside a grid cell. But, they do not need to be constant over the whole water column. In preliminary experiments, we were in fact using a sinking speed that was increasing with depth for large POC. However, we feel that such an increase is not clearly demonstrated by observations and we thus decided to keep sinking speeds constant with depth. We discuss that in details in our response to John Dunne's review (see point 4 of our response to John Dunne). In the revised version of the manuscript, we clearly state that sinking speeds do not need to be constant with depth. However, they should be assumed constant within a grid cell.

**Results:**

**The good fit to observed POC profiles is really impressive at first sight. To make it even more convincing, you could explain what exactly sPOC and bPOC are ...** The definitions of sPOC and bPOC are now given in the description of PISCES in section 2.1. Furthermore, the nominal cutoff between the two size classes is also specified. We don't think that such a good fit can be obtained by properly tuning the parameters of small POC. Improving the fit requires to increase the sPOC concentrations in the interior of the ocean. The sinking speed of sPOC cannot be changed because it is constrained by its (small) size. Thus, the only remaining way is to decrease remineralization rates. However, in the standard version of PISCES, remineralization rates are by definition constant over the global ocean (except for the effect of temperature). Thus, decreasing the remineralization rates at depth necessarily implies to decrease them also in the upper ocean. Based on the `RC` experiment, remineralization rates have to be reduced by one to two orders of magnitude. In the upper ocean, this would result in an increase in sPOC by about

[Figure]

Figure 1: Comparison between POC fluxes $(\mathrm{mg\,C/m^2/d})$ estimated from sediment traps and modeled fluxes. Black and red triangles denote the `NoRC` and the `RC` experiments respectively. Data are from Dunne et al. (2005), Gehlen et al. (2006) and Le Moigne et al. (2013).

the same magnitude (at least if the model is not too non-linear). sPOC would have concentrations comparable to DOC which is not observed.

**Figure 7, lower panels: Please choose a different colour scale. It is very difficult to see the dots for observations.** We have redrawn that figure in which the observations should be easier to see now. Furthermore, we now provide with the manuscript supplementary materials in which we provide a figure similar to Figure 3 of Gehlen et al. (2006). This figure is quoted now in the result section of the manuscript (Figure S2). The figure is provided in this response (see Figure 1).

**Figure 8 looks very impressive. Did you also compare against data set by Seiter et al?** In the submitted version of the manuscript, we did not compare our model result to the quoted data set. We have tried to retrieve those data but they don't seem to be publicly available, at least the dataset which has been used to draw Figure 7 in Seiter et al. (2005). It would be of great value if this dataset could be made available to the community.

**Figure 9 and p. 12, lines 13: How do you compute nu from the RC model? As the model's assumption was nu=1 ...** In our model, we make the assumption that $\nu$ is equal to 1 in freshly produced organic matter. However, we do not constrain the lability distribution in POC to follow a gamma distribution with a shape factor of 1. The lability distribution evolves with time and space in response to sources and sinks of POC. In fact, the $\nu$ displayed on Figure 9 is the apparent $\nu$ of the lability

distribution. It is computed from the actual lability distribution of POC using the mean and the variance. In a gamma distribution, $\nu$ is equal to the square of the mean divided by the variance. As stated by the reviewer, Boudreau and Ruddick (1991) found quite huge variations of $a$ (see Figure 6 in Boudreau and Ruddick (1991)). However, these values were obtained from sediment data. They explain these huge variations by the different sedimentation velocities: As sedimentation velocities decrease, the fast decaying part of the organic matter is removed before being incorporated in the historic part of the sediments. Thus, values of $a$ being deduced from sediment observations cannot be directly used in the oceanic domain. In the revised version of the manuscript, we describe the computation of the apparent $\nu$ in the legend of Figure 9 as well as in text on page 12.

**Discussion**

**The discussion repeats some things that were already presented in the methods description. Then, several other experiments (nu = 0.16, p12 ; 9 lability classes, modeled both implicitly and explicitly) ...** These additional experiments have been performed to test the robustness of our assumptions. The experiment with $\nu = 0.16$ has been designed to test if the value of the shape coefficient in the open ocean can be as low as some of the values inferred from sediment data. The experiment with 9 lability classes has been designed to make a comparison with a simulation in which the 9 lability classes are explicitly modeled. All these experiments support the discussion presented in the discussion section whose objective is to analyse the potential limitations of our study. In the result section, we rather show the potential impacts a variable lability of organic matter could have on the ocean carbon cycle. In that sense, the additional experiments do not bring any support to those objectives of the result section. That is why we think these experiments should be presented and discussed in the discussion section. As mentioned above, the lower incomplete gamma function is computed in the model using the algorithm proposed by MacLeod (1989). This algorithm is very efficient and the additional cost is minimal. Furthermore, the incomplete gamma function is only computed at the beginning of the simulation to define the lability classes (the $\bar{k}_i$ classes).

**Given the quite large uncertainties associated with the parameterization of the iron cycle, I find a comparison against dissolved iron concentrations quite ambitious ...** We would like to thank the reviewer for this comment. In fact, the legend of Figure 11 in the submitted version of the manuscript is incorrect. It displays the average of the difference between the simulated iron distributions and the observed iron distributions for both experiments. In the revised version of the manuscript, we have changed the caption of this figure. For nitrate, as explained in the submitted version of the manuscript, changes are much smaller. For oxygen, differences are more significant but do not exhibit the same vertical distribution as for iron. The anomalies occur deeper in the water column and extend over a much larger vertical extent. This is not surprising because oxygen and iron experience different processes. In particular, in the interior of the ocean, the iron distribution is quite strongly constrained by scavenging and the ligands distribution (at least in our model) which do not affect oxygen. In response to John Dunne's comment (reviewer 1), the discussion on oxygen and nitrate has been slightly more detailed in the revised version of the manuscript.

**There have been two recent papers that attempt to simulate the degradation of particles in local or global models ...** We would like to thank the reviewer because we were not aware of the first study. In that study by Jokulsdottir and Archer (2016), a detailed model of particles is being designed. It relies on a stochastic lagrangian description of particles. Many processes are explicitly described and modeled including aggregation, fragmentation by zooplankton, ingestion by zooplankton, bacterial degradation. A ballast effect is also represented through its impact on the density of the aggregates. This detailed model also makes the assumption that aggregates are composed of different compounds with distinct labilities, in a manner similar to our approach. As a consequence, when the aggregates age, their degradation rate decreases. However, in their study, they don't explore the impacts of this varying lability on the POC distribution and on the POC fluxes. This prevents a direct comparison to our approach and results. Their model is also very complex and its implementation in a classical biogeochemical model should prove to be quite difficult which is mentioned in the conclusions of their study. They suggest that a required step is to significantly simplify their model. They envision to use their model or a simplified version of it to study the transfer efficiency of POC in the mesopelagic domain and potentially to analyse the variations of this transfer efficiency as a function of environmental and biogeochemical parameters. If this study is done, it should prove to be very useful to better understand and constrain the processes that control the fate of POC in the ocean.

The reviewer mentions a second study performed by DeVries et al. (2014). In their work, they develop a more classical model of particles that solves the evolution of the size distribution of particles with depth. They explore the sensitivity of their model to different parameters: size distribution in the upper ocean, sinking speed, mass-volume relationship, ... Their conclusion is that observed fluxes can only be reproduced when a protective ballast is modeled. This is not very different to some extent to the conclusions of previous studies (Armstrong et al., 2001; Francois et al., 2002; Gehlen et al., 2006). They also mention that an alternative explanation to the ballast hypothesis could be the existence of more refractory compounds in organic matter such as polyphosphates (Diaz et al., 2008). This alternative explanation is compatible with our study which assumes that organic matter is made of various compounds of which some are quite recalcitrant. However, two points are missing in their study which prevents insightful comparison with our work. First, they don't display and analyse the POC distribution in the water column. Second, the contribution of small particles to the total flux in the deep ocean is very small in DeVries et al. (2014), much smaller than what is suggested by observations (Durkin et al., 2015). Unfortunately, the authors don't show this contribution for their ballast experiment. Thus, it is impossible to evaluate if their improved parameterization improves the contribution of small particles to the total flux of POC in the interior of the ocean.

In the revised version of our manuscript, we now quote these two studies and briefly discuss their results and their implications for our work.

**Specific comments**

**p1, line 13: The particles flux profile introduced by Suess is not an exponential one ...** The reviewer is correct. We have corrected this in the revised version of the

mansucript and we quote now proper references: (Lutz et al., 2002; Boyd and Trull, 2007).

**p6, Eqn(7) What are the P-term and the S-term exactly?** This point was also mentioned in the reviewer's main concerns. Some terms were not properly defined in our submitted version of the manuscript. This is now the case in the revised version. P is the production of POC and S is the sink of POC.

**p6, line 20 "Assuming constant sources and sinks over each grid cell" constant over what? Time?** This point was also mentioned by the reviewer in his main concerns. These lines have been extended to better explain the method (see our response to the reviewer's main concerns). We assumed that sources and sinks do not vary spatially within a grid cell. In other words, we assume that sources and sinks are homogeneous within each grid cell of the model. But they do change with time. The text is now: *"Assuming spatially constant sources, sinks and sinking speed within a grid cell (i.e., sources, sinks and sinking speeds are homogeneous within each grid cell), this system can be solved analytically over each grid cell."*

**Table 1: The number of lability classes refers to each POC compartment, correct?** We have remade Table 1 which now displays more information. The values of $k_i$ are given at the bottom of page 5 of the submitted version of the manuscript.

**Table 2: It seems to me as if headers (RC, noRC) are swapped. Please comment on the different acronyms ...** Caption of Table 2 has be extended to define now the acronyms. And the headers have been swapped.

**Table 3: What exactly is POC production - production of fecal pellets ? What is Phi? ...** Caption of Table 6 has also been significantly extended to better describe the different terms displayed in the table.

**References**

Armstrong, R., Lee, C., Hedges, J., Honjo, S., and Wakeham, S.: A new, mechanistic model for organic carbon fluxes in the ocean based on the quantitative association of POC with ballast minerals, Deep-Sea Res. Pt II, 49, 219–236, doi:10.1016/S0967-0645(01)00101-1, 2001.

Boudreau, B. P. and Ruddick, B. R.: On a reactive continuum representation of organic matter diagenesis, Am. J. Sci., 291, 507–538, doi:10.2475/ajs.291.5.507, 1991.

Boudreau, B. P., Arnosti, C., Jørgensen, B. B., and Canfield, D. E.: Comment on "Physical Model for the Decay and Preservation of Marine Organic Carbon", Science, 319, 1616–1616, doi:10.1126/science.1148589, 2008.

Boyd, P. W. and Trull, T. W.: Understanding the Export of Biogenic Particles in Oceanic Waters: Is There Consensus?, Progress in Oceanography, 72, 276–312, doi:10.1016/j.pocean.2006.10.007, 2007.

DeVries, T., Liang, J.-H., and Deutsch, C.: A mechanistic particle flux model applied to the oceanic phosphorus cycle, Biogeosci., 11, 5381–5398, doi:10.5194/bg-11-5381-2014, URL http://www.biogeosciences.net/11/5381/2014/, 2014.

Diaz, J., Ingall, E., Benitez-Nelson, C., Paterson, D., de Jonge, M. D., McNulty, I., and Brandes, J. A.: Marine Polyphosphate: A Key Player in Geologic Phosphorus Sequestration, Science, 320, 652–655, doi:10.1126/science.1151751, 2008.

Dunne, J. P., Armstrong, R. A., Gnanadesikan, A., and Sarmiento, J. L.: Empirical and mechanistic models for the particle export ratio, Global Biogeochem. Cy., 19, doi:10.1029/2004GB002390, 2005.

Durkin, C. A., Estapa, M. L., and Buesseler, K. O.: Observations of carbon export by small sinking particles in the upper mesopelagic, Marine Chemistry, 175, 72–81, doi:10.1016/j.marchem.2015.02.011, 2015.

Francois, R., Honjo, S., Krishfield, R., and Manganini, S.: Factors controlling the flux of organic carbon to the bathypelagic zone of the ocean, Global Biogeochem. Cy., 16, 1087, doi:10.1029/2001GB001722, 2002.

Gehlen, M., Bopp, L., Emprin, N., Aumont, O., Heinze, C., and Ragueneau, O.: Reconciling surface ocean productivity, export fluxes and sediment composition in a global biogeochemical ocean model, Biogeosci., 3, 521–537, doi:10.5194/bg-3-521-2006, 2006.

Jokulsdottir, T. and Archer, D.: A Stochastic, Lagrangian Model of Sinking Biogenic Aggregates in the Ocean (SLAMS 1.0): Model Formulation, Validation and Sensitivity, Geosci. Model Dev., 9, 1455–1476, doi:10.5194/gmd-9-1455-2016, 2016.

Le Moigne, F. A. C., Henson, S. A., Sanders, R. J., and Madsen, E.: Global database of surface ocean particulate organic carbon export fluxes diagnosed from the $^{234}$Th technique, Earth System Science Data, 5, 295–304, doi:10.5194/essd-5-295-2013, 2013.

Lutz, M., Dunbar, R., and Caldeira, K.: Regional Variability in the Vertical Flux of Particulate Organic Carbon in the Ocean Interior, Global Biogeochemical Cycles, 16, 11–1, doi:10.1029/2000GB001383, 2002.

MacLeod, A.: Algorithm AS 245, A Robust and Reliable Algorithm for the Logarithm of the Gamma Function, Applied Statistics, 38, 397–402, 1989.

Seiter, K., Hensen, C., and Zabel, M.: Benthic carbon mineralization on a global scale, Global Biogeochem. Cy., 19, doi:10.1029/2004GB002225, 2005.

---

## Author Response (AR2)

IPSL/IRD/LOCEAN
4 place Jussieu, F75252 Paris, France

February 17, 2017

Arne Winguth,
Editor, Biogeosciences

Dear Prof. Arne Winguth,

As requested, we resubmit a revised version of the manuscript entitled "A reactivity continuum of particulate organic matter in a global ocean biogeochemical model" prepared by Olivier Aumont, Marco van Hulten, Matthieu Roy-Barman, Jean-Claude Dutay, Christian Eth and Marion Gehlen. We have thoroughly revised this manuscript following the reviewers' comments. We have joined to this letter detailed answers to Reviewers' comments.

Best wishes,

Olivier Aumont

**Response to Reviewer 1: John Dunne**

We would like to thank again John Dunne for his very constructive review which helped to improve our first submitted version of the manuscript.

**Response to Reviewer 2**

Reviewer 2 suggests to improve the model description. Following his comments, we tried to substantially expend the description of the new POC parameterization with regard to the first submitted version of the manuscript. However, some aspects seem to need some more explanations. We have tried to clarify our description following his new suggestions.

**In some cases, units can be very helpful for the reader ...** Following the reviewer's suggestions, we have tried to indicate the units for every variable that is introduced in our model description.

**It would, to my opinion, also be helpful to distinguish more clearly between "k", "overbar(k)", lambda, lambda\*, ... In fact, I would suggest to provide a table that explains all these parameters, their definitions, units, and values (if applicable).** We agree with the reviewer that a table could help to clarify the meaning of the different parameters. Following his suggestion, we have added a table which lists the different parameters, their meanings, their units and their values.

**Probably much of my initial confusion during the first review came from the fact that the title of the paper is "A reactivity continuum of particulate organic matter", whereas the authors seem to have a multi-G model (Eqn. 2 discretized into 15 lability classes for both sPOC and bPOC), with subgrid scale parameterizations via the Gamma function within each lability class. Correct? If this is the case, the title is a bit misleading.** The reviewer is mostly right here. We describe the lability distribution of newly produced organic matter using a continuous gamma function. Then, the lability distribution is discretized along the lability space which is equivalent to using a multi-G model. The title is thus a little bit misleading. We have changed the title of the manuscript to "Variable reactivity of particulate organic matter in a global ocean biogeochemical model". As a consequence, we do not refer anymore in the title to the reactivity continuum.

**Eqn. 8 is still difficult to understand for me: are these equations solved within each vertical grid box? More general, the interplay between vertical and lability discretization is a bit unclear to me. How does sinking exactly impact the lability distribution?** The system of equations is solved within each grid box. This is a typical system in a Lagrangian framework. The first equation relates time and space based on the sinking speed. This is where sinking speed comes into play in the formulation. To describe more clearly the algorithm that is being used in the model, we start from the bottom of the mixed layer. For each lability class, we compute from the first equation of the system the time the particles spend in a grid cell. Using that duration, we compute the impact of the different sources and sinks described in the second equation of the system. We end up with the new

lability distribution at the bottom of the considered grid cell and we iterate that procedure over the vertical domain. This procedure is described quite extensively at the top of page 7 of the revised version of the manuscript.

**Further, with regard to the last term of Eqn. 8 (second line), i.e. k_i f(T) ... : how does this term connect to the later statement (p. 7, line 11-12): "A mean k is inferred from that lability distribution which is then used in PISCES to model the decomposition of POC"? Is POC in each class decaying, or do you compute an average k, and then dissolve average POC from it?** Using the system of equation 8, we can compute the lability distributions for small POC and large POC in every grid cell of the model. From those distributions, we compute the mean remineralization rates of small POC and large POC which is called $\bar{k}$. Thus, for every grid cell, we have a mean remineralization rate for small POC and another mean remineralization rate for large POC. The decay of both pools is then computed as $-\bar{k}_{sPOC}sPOC$ and $-\bar{k}_{bPOC}bPOC$ for respectively small POC and large POC. We have slightly rewritten lines 11 and 12 of page 7 to make it clearer.

**Specific comments**

**p. 2, lines 24-25 "POC concentrations tend to be strongly underestimated in the deep ocean, which suggests that an excessive loss of POC is predicted in the mesopelagic zone." - Here a reference would be good to motivate why it could be helpful to include the new processes in any model.** We already quote two references here: Dutay et al. (2009, 2015).

**Eqn. 9 Is the mixed layer the only domain where mixing is taken into account for lability?** Yes, that's the only domain where vertical mixing is taken into account for lability. We have added the following sentence on line 19 of page 7 "The mixed layer is the only domain where the lability distribution is impacted by vertical mixing".

**p. 6, line 25 "each pool" - each lability class?** Yes, the reviewer is right. We have changed pool to lability class in the revised version of the manuscript.

**p. 7, line 33: "rate parameter a" - better "lifetime parameter a"?** Following the reviewer's suggestion, we have changed rate parameter a to lifetime parameter a.

**p. 10, lines 30-32: Could this indicate either a too slow decrease of lability with depth, or the lack on an increase of particle sinking speed with depth?** A too slow decrease of the lability with depth would lead to fresher materials in the mesopelagic domain and in the deep ocean. The consequence would be a too rapid decrease in POC in the mesopelagic domain and thus, a too low export at 2000m which is not what we simulate since our predicted export at this depth horizon is overestimated. And it would also lead to a too low export to the sediment which is not what the model simulates. The second hypothesis suggested by the reviewer may be possible. In fact, assuming that the sinking speed of large particles is increasing with depth would allow to decrease the sinking flux at 2000m without necessarily decreasing the export to the sediments.

**Figure 6, caption: what is GOC?** Sorry, it is a typo. This variable name has been changed to bPOC.

**Figure 10: How was "b" computed? In particular: over which vertical domain?** b has been computed by fitting a power law function to the modeled profiles of total particulate carbon as simulated in the different model experiments between the bottom of the photic zone and 1000m. The caption of Figure 10 has been modified to include that explanation "b has been determined by fitting a power-law function to the modeled vertical profiles of total particulate carbon between the bottom of the photic zone and 1000m"

[revised manuscript text omitted]

10  lability are listed in Table 1. As in Boudreau and Ruddick (1991), the distribution of reactivities of newly produced POC is represented by a gamma distribution:

$$g(k,0) = \frac{g_0 k^{\nu-1} e^{-ak}}{\Gamma(\nu)}, \tag{1}$$

where $\nu$ describes the shape of the distribution near $k = 0$, and $a$ (days) is the average life time of the more reactive components of POC. The corresponding cumulative distribution function (CDF) is defined as:

15  $$\mathcal{G}(k,0) = \frac{\gamma(\nu, ak)}{\Gamma(\nu)} = \frac{\int_0^{ak} x^{\nu-1} e^{-x} \mathrm{d}x}{\int_0^\infty x^{\nu-1} e^{-x} \mathrm{d}x}, \tag{2}$$

where $\gamma(\nu, ak)$ is the lower incomplete gamma function. To get the time-evolved distribution, we assume first-order decay for each lability class:

$$g(k,t) = \frac{g_0 k^{\nu-1} e^{-(a+f(T)t)k}}{\Gamma(\nu)}, \tag{3}$$

where $f(T)$ is a function of temperature $T$ (°C). As in the standard version of PISCES, the dependency to temperature

20  corresponds to a $Q_{10}$ of 1.9. In that equation, the effect of temperature on the distribution is equivalent to defining a pseudo time variable $t^\star = f(T)t$.

In a closed system with a constant temperature $T$, the mean remineralisation rate of POC decreases with time and is described by a simple function of the pseudo time variable $t^\star$ (Boudreau and Ruddick, 1991; Boudreau et al., 2008):

25  $$\frac{\mathrm{d}POC}{\mathrm{d}t^\star} = -\frac{\nu}{a+t^\star} POC. \tag{4}$$

Unfortunately, in open systems such as the ocean, the RC model cannot be used in its continuous form since transport, production and consumption of organic matter alter the shape of the distribution. As a consequence, this distribution can significantly deviate from the initial gamma distribution. An option would be to model the moments of the reactivity distribution following an approach similar to what is done for instance in the atmosphere

30  for aerosols (Milbrandt and Yau, 2005) or in the ocean for traits (Merico et al., 2009). However, the set of moment

equations should be closed using a moment closure approximation to truncate the system at a certain order. Since the distribution can deviate to a non-specific form, the number of moments that should be tracked becomes very large (over ten moments based on some preliminary tests we performed) which makes that method computationally inefficient. Instead, the distribution is discretised by separating both small and large POC into a finite number of
5   pools having degradation constants that are equally spaced in the natural logarithmic transform of the reactivity space. In this study, we have arbitrarily set the smallest and largest boundaries of the lability classes in the reactivity space to respectively  $\tilde{k}/1000$ and $10\tilde{k}$, where $\tilde{k}$ $(\mathrm{d}^{-1})$ is the mean degradation rate of freshly produced POC. Each boundary $k_i$ $(\mathrm{d}^{-1})$ in the lability space is computed as:

$$k_i = \frac{1}{1000}(1 \times 10^4)^{\frac{i-2}{n-2}}\tilde{k} \quad \text{for} \quad i = 2, n, \tag{5}$$

10   where $n$ is the number of lability classes. The first $k_1$ and last $k_{n+1}$ boundaries are defined as:

$$k_1 = 0 \quad \text{and} \quad k_{n+1} = +\infty \tag{6}$$

The fraction $\mathcal{G}(\bar{k}_i, 0)$ and the mean degradation rate $\bar{k}_i$ $(\mathrm{d}^{-1})$ of POC having degradation constants between $k_i$ and $k_{i+1}$ are:

$$\begin{aligned} \mathcal{G}(\bar{k}_i, 0) &= \frac{\gamma(\nu, ak_{i+1}) - \gamma(\nu, ak_i)}{\Gamma(\nu)} \\ \bar{k}_i &= \frac{\gamma(\nu+1, ak_{i+1}) - \gamma(\nu+1, ak_i)}{\Gamma(\nu)} \quad \text{for} \quad i = 1, n \end{aligned} \tag{7}$$

15   We thus use a multi-G model to simulate the reactivity continuum. An identical approach has been used by Dale et al. (2015) to model the degradation of the organic matter in the sediments. Based on experiments performed with a 1-D model, the number of pools has been set to 15 for both small and large POC which results in a less than $1\%$ error relative to the exact solution. Gamma functions are computed in our model using the algorithm proposed by MacLeod (1989).

20     An explicit representation of the reactivity of the organic matter would require to have 30 distinct pools, which would more than double the number of variables in  PISCES (24 tracers including large and small POC). This would thus considerably increase the computing cost of the model. To overcome that problem, we made a rather strong assumption: We postulated that the lability distribution of POC is insensitive to ocean transport and is only modified by sinking, by the biological sources and sinks, and by vertical mixing in the mixed layer. This assumption
25   is further discussed in the discussion section of this study. The problem is thus reduced to a 1-D framework and the vertical distribution of each  lability class can be iteratively solved starting from the base of the mixed layer. In a lagrangian framework, the system to be solved is:

$$\begin{cases} \dfrac{\mathrm{d}z}{\mathrm{d}t} = w_{\mathrm{POC}} \\ \dfrac{\mathrm{d}\mathcal{G}(\bar{k}_i, t)POC}{\mathrm{d}t} = \mathcal{G}(\bar{k}_i, 0)P_{\mathrm{POC}} - \mathcal{G}(\bar{k}_i, t)S_{\mathrm{POC}} - \bar{k}_i f(T)\mathcal{G}(\bar{k}_i, t)POC \\ \mathcal{G}(\bar{k}_i, t) = \mathcal{G}_{\mathrm{mxl}}(\bar{k}_i, t) \quad \text{if} \quad z = z_{\mathrm{mxl}} \end{cases} \tag{8}$$

where $z$ (m) is positive downwards, $P_{\text{POC}}$ (mol m$^{-3}$ d$^{-1}$) denotes the production of particulate organic carbon, $S_{\text{POC}}$ (mol m$^{-3}$ d$^{-1}$) represents the sinks of POC, $w_{\text{POC}}$ (m d$^{-1}$) the settling velocity of POC, $z_{\text{mxl}}$ (m) the depth of the bottom of the mixed layer, and $\mathcal{G}(\bar{k}_i, t)$ $\mathcal{G}(\bar{k}_i, t)$ corresponds to the mass fraction of POC with a decay rate $\bar{k}_i$ at time $t$. Sources of POC ($P_{\text{POC}}$) are mainly mortality of phytoplankton and zooplankton, exudation of fecal pellets

5 and coagulation of phytoplankton cells. All these processes are assumed to produce fresh new POC characterized by the lability distribution $\mathcal{G}(\bar{k}_i, 0)$. Sinks of POC essentially correspond to grazing by zooplankton. We assume here that grazing does not depend on lability, i.e. each lability class is impacted proportionately with its relative contribution $\mathcal{G}(\bar{k}_i, t)$. The sinking speed $w_{\text{POC}}$ does not need to be constant with depth. Assuming spatially constant sources, sinks and sinking speed within a grid cell (i.e., sources, sinks and sinking speeds are homogeneous within each grid

10 cell), this system can be solved analytically over each grid cell. The vertical distribution of each lability class (with a decay rate $\bar{k}_i$) is then computed iteratively over the water column starting from the mixed layer. A mean $\bar{k}$ (d$^{-1}$) is inferred from that lability distribution which is then used in  PISCES to model the decomposition of POC (the sink term of POC due to degradation by bacterial activity is computed as $-\bar{k}POC$; For more information on the description of POC in PISCES, please refer to Aumont et al. (2015)). As a consequence, $\bar{k}$ displays both vertical

15 and horizontal variations and is thus a 3-D field. Of course, this computation is done independently for small and large POC.

The solution of system (8) requires to know the distribution at the bottom of the mixed layer $G_{\text{mxl}}(k, t)$. In the mixed layer, ocean dynamics, especially vertical mixing, cannot be neglected. Since vertical mixing is strong, tracers in the mixed layer, including the reactivity distribution, can be considered homogeneous. Using that assumption,

20 the mean reactivity distribution $\mathcal{G}_{\text{mxl}}(\bar{k}_i, t)$ can be computed by integrating Equation 8 over the mixed layer:

$$\mathcal{G}_{\text{mxl}}(\bar{k}_i, t) = \frac{\int_0^{z_{\text{mxl}}} \mathcal{G}(\bar{k}_i, 0) P_{\text{POC}} \mathrm{d}z}{\int_0^{z_{\text{mxl}}} (\bar{k}_i f(T_{\text{mxl}}) POC + S_{\text{POC}}) \mathrm{d}z + w_{\text{POC}} POC(z = z_{\text{mxl}})} \tag{9}$$

[revised manuscript text omitted]

**Figure 11.** Iron distribution (nmol Fe/L) in the Pacific ocean. The left panel shows the zonal average of the difference between the `RC` and the `NoRC` experiments. The isolines display the zonally averaged iron distribution as simulated in the `NoRC` experiment. The right panel shows the vertical profiles of the difference between the simulated iron distributions (red solid line: the `NoRC` experiment; Green solid line: the `RC` experiment) and observed iron from Tagliabue et al. (2012). Differences have been averaged over the Pacific Ocean. The light shaded areas display the standard deviation of the differences (red: the `NoRC` experiment; Green: the `RC` experiment).

**Table 1.** Parameters and variables with associated units. When two values are specified in the value column, they refer to respectively sPOC and bPOC.

| Symbol | Description | Units | Value |
|---|---|---|---|
| $\nu$ | Shape parameter of the gamma distribution | – | 1.0 |
| $a$ | Average lifetime of the more reactive component | days | See Table 2 |
| $t^\star$ | Pseudo time variable | d | $f(T)t$ |
| $T$ | Temperature | °C | Variable |
| $POC$ | Particulate Organic Carbon | $\mathrm{mmol\,m^{-3}}$ | Variable |
| $sPOC$ | Small Particulate Organic Carbon | $\mathrm{mmol\,m^{-3}}$ | Variable |
| $bPOC$ | Large Particulate Organic Carbon | $\mathrm{mmol\,m^{-3}}$ | Variable |
| $\tilde{k}$ | Mean degradation rate of fresh POC | $\mathrm{d^{-1}}$ | See Table 2 |
| $k_i$ | Boundary in the lability space | $\mathrm{d^{-1}}$ | From Eq. 5 |
| $\mathcal{G}(\bar{k}_i,t)$ | Fraction of POC with degradation rate $\in [k_i, k_{i+1}[$ | – | Variable |
| $\mathcal{G}_{mxl}(\bar{k}_i,t)$ | Mean $\mathcal{G}(\bar{k}_i,t)$ in the mixed layer | – | From Eq. 9 |
| $\bar{k}_i$ | Mean degradation rate of $\mathcal{G}(\bar{k}_i,0)$ | day | From Eq. 7 |
| $P_{POC}$ | Production of POC | $\mathrm{mol\,m^{-3}\,d^{-1}}$ | Variable |
| $S_{POC}$ | Sink of POC | $\mathrm{mol\,m^{-3}\,d^{-1}}$ | Variable |
| $w_{POC}$ | Settling speed of POC | $\mathrm{m\,d^{-1}}$ | 2, 50 |
| $z_{mxl}$ | Depth of the mixed layer | m | Variable |
| $\bar{k}$ | Mean degradation rate of POC | $\mathrm{d^{-1}}$ | Variable |
| $T_{mxl}$ | Mean temperature in the mixed layer | °C | Variable |
| $f(T)$ | Temperature dependency of degradation rate | – | Variable |
| $b$ | Exponent of the Martin et al. (1987) power-law function | – | Variable |

**Table 2.** Model simulations.

| Experiment | #lability classes[a] | $\lambda^\star_{POC}$ $\tilde{k}$ | $a$ |
|---|---|---|---|
| noRC | 1 | 0.025 d$^{-1}$ | |
| RC | 15 | 0.035 d$^{-1}$[b] | 28.57 d |

[a] For each POC compartment (sPOC and bPOC)

[b] Remineralization rate of freshly produced POC.

**Table 3.** Statistical model–data comparison of the two simulations for the whole particle dataset displayed in Figure 3. The upper ocean has been excluded from the analysis (defined as less than 200m). RMSE, RI, and MEF are respectively the root mean square error, the reliability index and the modeling efficiency (see text for more information).

|      | NoRC | RC   |
| ---- | ---- | ---- |
| $n$  | 2656 | 2656 |
| $r$  | 0.7  | 0.77 |
| RMSE | 0.14 | 0.08 |
| Bias | 0.12 | 0.02 |
| RI   | 4.9  | 1.6  |
| MEF  | -0.6 | 0.71 |

**Table 4.** Primary production and fluxes of POC ($\Phi$) integrated over the global ocean for the different simulations, and estimates based on observations. All numbers are in units of PgC/yr of carbon. The numbers in parentheses denote the fluxes due to small POC. The photic depth is defined as the depth at which photosynthetic available radiation equals 1% of the value at the ocean surface. $\Phi_{sed}(z > 1000\,\text{m})$ is the flux of POC to the sediments whose depth is deeper than 1000 m.

| Simulation | primary production | $\Phi_{\text{photic depth}}$ | $\Phi(z = 2000\,\text{m})$ | $\Phi_{\text{sed}}(z > 1000\,\text{m})$ |
| --- | --- | --- | --- | --- |
| NoRC | 52 | 9 (3.6) | 0.56 (0.01) | 0.19 (0) |
| RC | 41 | 8.1 (3.6) | 0.81 (0.17) | 0.42 (0.11) |
| Published estimates | 40–60[a] | 4–12[b] | 0.33–0.66[c] | 0.5–0.9[d] |

[a] from Carr et al. (2006);

[b] from Siegel et al. (2014), Laws et al. (2000), Lutz et al. (2007), Dunne et al. (2007) and Henson et al. (2011);

[c] from Henson et al. (2012b) and Guidi et al. (2015);

[d] from Jahnke (1996) and Seiter et al. (2005);